# Nucleoplasmic Lamin A/C controls replication fork restart upon stress by modulating local H3K9me3 and ADP-ribosylation levels

Veronica Cherdyntseva [1], Joanna Paulson[2], Daniel González-Acosta [1], Patricia Ubieto-Capella [1], Melani Rodrigues [1], Moses Aouami[1], Selin Adakli [1], Jean-Philippe Gagné [3,4], Collin Bakker[2], Guy G. Poirier[3,4], Nitika Taneja [2] & Massimo Lopes [1] ✉

Mild replication interference is a consolidated strategy for cancer chemotherapy. Tolerance to mild replication stress (RS) relies on active fork slowing, mediated by transient fork reversal and RECQ1-assisted restart, and modulated by PARP1 and nuclear architectural components via yet-elusive mechanisms. We combined acute protein inactivation with cell biology and single-molecule approaches to investigate the role of Lamin A/C upon mild RS. We found that Lamin A/C dynamically interacts with replication factories throughout the nucleus and, together with its nucleoplasmic partner LAP2α, is required to induce active fork slowing and maintain chromosome stability upon mild genotoxic treatments. Inactivating nucleoplasmic Lamin A/C reduces poly-ADP-ribosylation (PAR) levels at nascent DNA, triggering deregulated RECQ1-mediated restart of reversed forks. Moreover, we found that the heterochromatin mark H3K9me3, previously reported at stalled forks, also accumulates in response to mild RS. H3K9me3 accumulation requires Lamin A/C, which prevents its premature removal by the histone demethylase JMJD1A/KDM3A. H3K9me3 loss per se phenocopies Lamin A/C inactivation, reducing PAR levels and deregulating fork restart by RECQ1. Hence, nucleoplasmic Lamin A/C, H3K9me3 and PARylation levels are crucial, mechanistically linked modulators of fork dynamics upon mild RS, with important implications for chemotherapy response and for Lamin A/C dysfunction in human disease.

DNA replication is an essential process for the transmission of genetic information during somatic cell division. Completeness and accuracy of this process are essential to maintain proper genome duplication across generations. DNA replication is challenged by multiple endogenous and exogenous sources of genotoxic stress – collectively reported as replication stress (RS) – including various types of DNA lesions, limited nucleotide levels and interference with transcription[1]. The mechanisms by which these obstacles are tackled to ensure genome stability are highly relevant to avoid cancer onset and contribute to known mechanisms of resistance of cancer cells to commonly used

[1]Institute of Molecular Cancer Research, University of Zurich, Zurich, Switzerland. [2]Department of Molecular Genetics, Oncode Institute, Erasmus University Medical Center, Erasmus MC Cancer Institute, Rotterdam, the Netherlands. [3]Department of Molecular Biology, Medical Biochemistry and Pathology, Université Laval, Quebec City, Canada. [4]CHU de Quebec Research Center, CHUL Pavilion, Oncology Axis, Quebec City, Canada. ✉e-mail: lopes@imcr.uzh.ch

chemotherapeutics[1,2]. Cells respond to RS by activating signaling pathways and replication fork protection mechanisms, which are best characterized upon conditions that block DNA synthesis and induce prolonged fork stalling. In fact, to limit toxicity in normal cells and arrest the uncontrolled proliferation of cancer cells, commonly used chemotherapeutic regimens are compatible with residual DNA synthesis. They rather trigger specialized mechanisms of RS tolerance that capitalize on replication fork plasticity, allowing DNA synthesis to continue in unfavorable conditions, albeit at a slower pace[3]. Even upon mild RS-inducing treatments, a high fraction of replication forks in human cells undergo transient remodeling into 4-way junctions, in a process known as replication fork reversal[4]. Reversed fork formation is modulated by a growing number of replication accessory factors, some of which have been previously implicated in classical DNA repair[5]. Also restart of reversed forks requires specialized enzymes, such as the RECQ1 helicase, which is controlled by its interaction with poly-ADP-ribose (PAR), synthesized by PARP1 on itself and additional targets[3,6,7]. An accurate balance of fork reversal and restart is crucial to mediate active fork slowing upon mild RS and to resist drug-induced conditions of RS, impacting on cancer therapy response in multiple tissues[2].

The DNA replication template is packed in nucleosomes containing specific epigenetic marks and is organized in a higher order chromatin structure, which determines the gene expression profile of each cell. Hence, removal and restoration of chromatin marks, as well as chromatin organization during DNA replication are of crucial importance to maintain cell identity[8,9]. RS adds complexity to this already challenging task, by affecting simultaneous and complete DNA synthesis on the two template strands and requiring spatiotemporally controlled access of specific accessory factors to replicating chromatin[10]. Deposition of specific histone variants and histone marks on replicated DNA are known to assist replication-coupled repair and maintain fork integrity[11,12]. Transient accumulation of heterochromatic marks – typically associated with silent chromatin – on newly replicated DNA was recently shown to mediate the cellular response to replication fork stalling, by limiting the access of DNA synthesis restart factors, such as Primpol[13]. Whether similar mechanisms assist the immediate response to mild and clinically relevant genotoxic treatments – affecting DNA synthesis without stalling replication forks – has remained elusive.

Besides the established role of chromatin organization in the RS response, several components of nuclear architecture and nuclear dynamics are also emerging as key players upon replication interference. Nuclear actin filaments were shown to assist repair of DNA breaks[14,15] and to relocate forks experiencing prolonged stalling[16]. More recently, the nuclear actin network was shown to protect the stability of stalled forks[17,18] and to modulate fork progression upon mild RS, by limiting recruitment of Primpol to transiently challenged forks[19]. Considering that replication fork remodeling appears to extend in the nucleus beyond the forks directly experiencing obstacles[20], these recent findings suggest that proper coordination of the RS response requires local and global control of nuclear architecture and 3D genome organization[21]. Regulated loading and unloading of cohesin –another key player in genome organization and nuclear dynamics – controls replication timing in the nucleus[21,22], but was also shown to support fork stability, progression and restart in different conditions of RS[23]. Based on this growing evidence, it seems likely that several additional components of nuclear architecture play pivotal roles upon RS, possibly impacting genome stability and cellular resistance to cancer chemotherapy.

Lamin A/C is another key component of nuclear architecture, best known for its role within the dense fibrillar network of intermediate filaments supporting structurally the nuclear membrane (lamina)[24]. Mutations impairing this structural function have profound consequences at cellular and organismic level, impacting the mechanical properties of the cells in specific tissues and leading to a heterogeneous set of diseases, collectively called laminopathies[25]. Although most cellular Lamin A/C is assembled in the nuclear lamina, a significant fraction of the protein resides in the nucleoplasm in a more soluble and less detectable form, bound to its specific nucleoplasmic partner LAP2α[26,27]. Nucleoplasmic Lamin A/C and LAP2α appear to modulate chromatin mobility in the nuclear interior[27,28]. Moreover, Lamin A/C was shown to interact directly with the histone lysine methyltransferase SUV39H1[29]; however, whether and how Lamin A/C-dependent modulation of chromatin organization affects gene expression and other nuclear functions is still elusive. Lamin A/C was previously investigated in the context of replication fork stalling and shown to promote fork restart, possibly by mediating efficient recruitment of ssDNA-binding proteins RPA and RAD51[30,31]. A role in RPA binding and recruitment to damaged chromatin was also recently proposed for LAP2α and reported to depend on PARP1[32]. Intriguingly, both Lamin A/C and LAP2α were recently found by proximity proteomics as interactors of PARP1[33], which – besides the established role at DNA breaks – is also recruited and activated at persistent discontinuities on nascent DNA[34,35]. A general limitation in previous investigations of Lamin A/C roles in DNA replication is the use of prolonged or permanent inactivation of the protein; considering the crucial structural functions of Lamin A/C, cells may need to adapt to its absence, promoting alternative mechanisms of nuclear organization and masking potentially interesting phenotypes. Overall, it is still unclear whether the role of Lamin A/C in DNA replication is limited to prolonged fork stalling or extends to mild RS conditions, whether this function entails the lamina or its nucleoplasmic pool, and whether it relates to the emerging links of Lamin A/C with chromatin organization.

Here, we show that Lamin A/C interacts with replication factories throughout the nucleus and that acute inactivation of Lamin A/C abolishes active fork slowing and increases genomic instability upon mild RS. These defects are phenocopied by genetic ablation of LAP2α and are linked to an impaired accumulation of PAR at replication forks, triggering the deregulated restart of reversed forks by the RECQ1 helicase. Moreover, we report that the accumulation of heterochromatic marks (H3K9me3) at replication forks is also detected upon mild genotoxic treatments, and their maintenance strictly requires Lamin A/C. Strikingly, impairing heterochromatic mark maintenance upon mild genotoxic treatments recapitulates all molecular defects observed upon Lamin A/C inactivation, suggesting that the control of chromatin compaction by nucleoplasmic Lamin A/C plays a key role in modulating PAR levels and RECQ1-mediated restart upon mild genotoxic stress.

## Results

### Lamin A/C interacts dynamically with replication factories throughout the nucleus

To investigate whether and where Lamin A/C establishes contacts with replication factories within the nucleus, we performed proximity ligation assays (PLA) between Lamin A/C and EdU, a thymidine analog that was briefly incorporated during DNA synthesis before cell preparation. We selected HCT116 colon cancer cells for these imaging analyses, especially due to the consistent round shape of their nuclei. Expectedly, Lamin A/C was mostly detectable by immunofluorescence (IF) at the nuclear periphery, but confocal imaging and 3D nuclei reconstruction showed numerous Lamin A/C-EdU PLA foci throughout the nucleus, suggesting that – besides the nuclear lamina - also low-abundant Lamin A/C within the nucleoplasm is in close contact with replication centers (Fig. 1a, b, Supplementary Fig. 1a; and Supplementary Video 1). As expected, *LMNA* downregulation decreased the number of Lamin A/C-EdU PLA foci, supporting the specificity of the antibody to target Lamin A/C (Supplementary Fig. 1b). To assess how Lamin A/C interaction with replication factories is affected upon mild conditions of RS, we exposed U2OS human osteosarcoma cells to mild

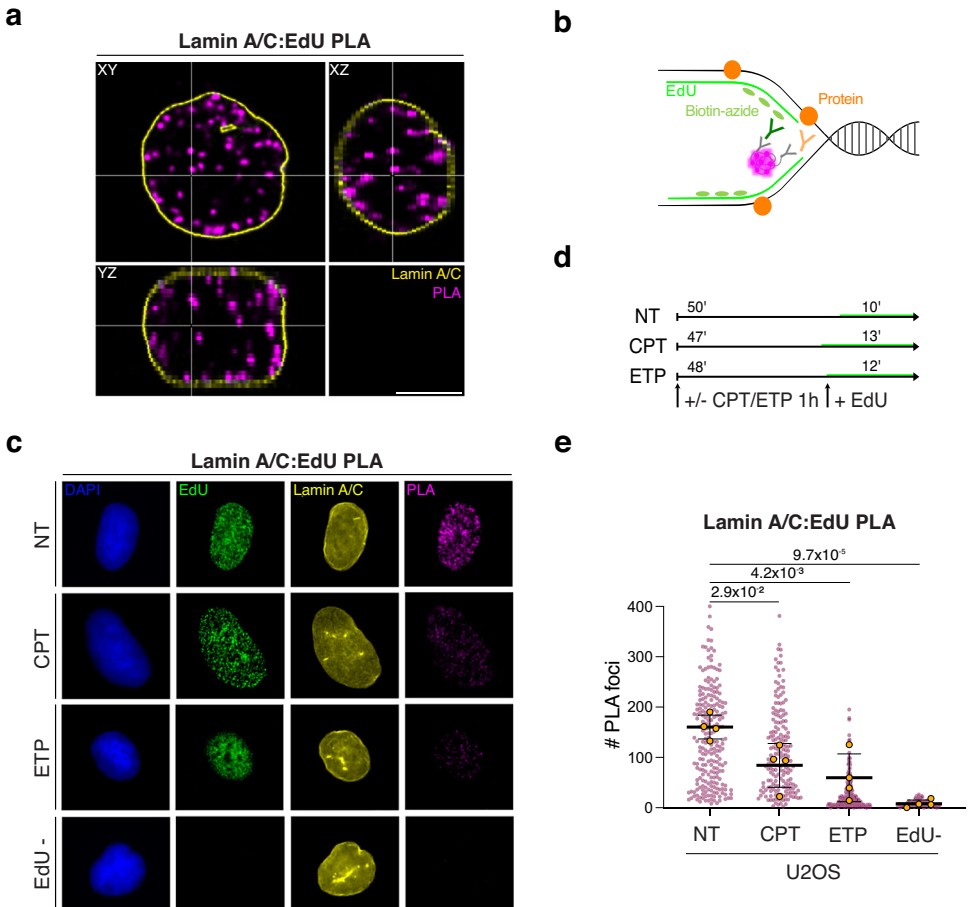

**Fig. 1 | Lamin A/C dynamically interacts with replication factories throughout the nucleus. a** Representative confocal microscopy image of HCT116 cells showing Lamin A/C in proximity to nascent DNA (EdU), detected by Lamin A/C:EdU PLA (magenta), Lamin A/C IF staining (yellow). While Lamin A/C is mainly detected at the nuclear periphery, its interaction with nascent DNA (EdU) is detected throughout the nucleus and in different axial perspectives (top left: XY view, bottom left: YZ view, top right: XZ view). Numerous comparable examples of this pattern were observed in two independent experiments. Scale bar, 5 μm. **b** Schematic representation of the Proximity Ligation assay used. EdU incorporation is followed by click chemistry with biotin-azide. Antibodies against the target protein and biotin are recognized by secondary antibodies carrying probes. When the target protein and EdU are in close proximity (< 40 nm), probes are ligated and amplified giving rise to a fluorescent PLA signal **c**. Experimental design for the IF/PLA experiment in (**c**). The duration of the EdU pulse is adapted to allow comparable incorporation of EdU despite the genotoxic treatments. **d** Representative U2OS nuclei (DAPI) – untreated or treated for 1 h with 100 nM CPT or 20 nM ETP–and stained for DNA synthesis (EdU), Lamin A/C and its physical proximity to nascent DNA (Lamin A/C:EdU PLA). Scale bar, 10 μm. **e** Quantification of Lamin A/C PLA signals from c. Signal was quantified in at least 100 EdU+ nuclei, in 4 independent experiments. EdU- cells are used as negative control. Yellow circles indicate the median for each experiment, while the black bar indicates the mean of the median values +/- SD. Statistical analysis was applied on the median values, using one-way ANOVA test with Bonferroni's *post hoc* correction.

treatment (100 nM) of camptothecin (CPT), a topoisomerase I inhibitor that induces marked fork slowing and remodeling, combined with low levels of replication-associated double-strand breaks (DSBs). Alternatively, we treated the cells with a mild dose (20 nM) of etoposide (ETP), the topoisomerase II inhibitor that also significantly impacts replication fork progression, without inducing detectable DSBs or checkpoint activation[6]. Similarly to HCT116 cells, widespread PLA signals were detectable by widefield IF imaging also in U2OS cells (Fig. 1c). Although the EdU labeling time was adjusted to allow comparable EdU incorporation upon each of the treatments, Lamin A/C-EdU PLA foci and intensities were significantly decreased upon both CPT and ETP treatments (Fig. 1d, e, and Supplementary Fig. 1c, d). This decrease may reflect Lamin A/C release from replication centers, or – as recently shown for other replisome components[36] – a switch to a more distant or dynamic interaction with nascent DNA due to fork remodeling during the RS response. LAP2α downregulation in U2OS cells decreased Lamin A/C-EdU PLA foci (Supplementary Fig. 1e–g), in line with its established role modulating other nucleoplasmic functions of Lamin A/C[27,28]. Overall, these data show a widespread

interaction of Lamin A/C with DNA replication centers throughout the nucleus, which is promoted by its nucleoplasmic interactor LAP2α and modulated upon mild RS, regardless of DSB formation.

## Acute depletion of Lamin A/C or its nucleoplasmic partner LAP2α abolishes active fork slowing upon stress

The roles of Lamin A/C in DNA replication or RS response have been explored so far upon prolonged or permanent (chronic) inactivation of the protein and in response to a complete replication blockage induced by nucleotide depletion[30,31]. To investigate functional roles of Lamin A/C upon mild RS, we took advantage of the AID2 technology[37] and developed an auxin-inducible degron HCT116 cell line targeting the LMNA protein by fusing the mAID2-mClover construct on the endogenous locus (*mAID2-LMNA*; Supplementary Fig. 2a). Hence, this cell line expresses a fluorescently-detectable protein (due to mClover) that can be efficiently degraded upon the addition to the culture media of 5-Ph-IAA (auxin analog, hereafter simply referred to as "auxin"). We isolated two clones (13 and 22) for further experiments in which we determined that full LMNA depletion is achieved 24 h after auxin

addition (Fig. 2a and Supplementary Fig. 2b–d). We confirmed that 24 h after auxin addition *mAID2-LMNA* HCT116 cells do not experience any marked alteration of their cell cycle progression, even when Lamin A/C depletion is prolonged up to 15 days (Supplementary Fig. 2c–g). Similarly, we found that marked *LMNA* downregulation can be achieved in HCT116 and U2OS cells 48 h after transfection with a specific siRNA (Fig. 2a, b) and that at this time point the cells do not yet display delayed cell cycle progression, impaired S phase entry or activation of the DNA damage response (Supplementary Fig. 2h, i). We used these controlled conditions of acute Lamin A/C depletion (Fig. 2a, b) to investigate replication fork progression at single-molecule level by DNA fiber spreading assays[38], providing cells with halogenated nucleotide analogs and with mild doses of ETP or CPT. These

treatments were previously shown to induce marked fork slowing and reversal, with no major impact on cell cycle progression and cell viability[6]. As expected, ETP and CPT markedly affected replication fork progression in both HCT116 and U2OS cells (Fig. 2c–e and Fig. 2f–h). However, the active fork slowing observed in these conditions is significantly rescued by acute or prolonged auxin-inducible Lamin A/C degradation in both *LMNA-mAID2-mClover* HCT116 clones (Fig. 2c–e and Supplementary 2j, k) or by *LMNA* downregulation in U2OS (Fig. 2f–h). Importantly, complete suppression of ETP/CPT-induced active fork slowing is also observed upon depletion of LAP2α (Fig. 2f–h), suggesting that this function specifically requires the nucleoplasmic fraction of Lamin A/C. To assess whether defective fork slowing upon Lamin A/C inactivation is linked to increased genomic

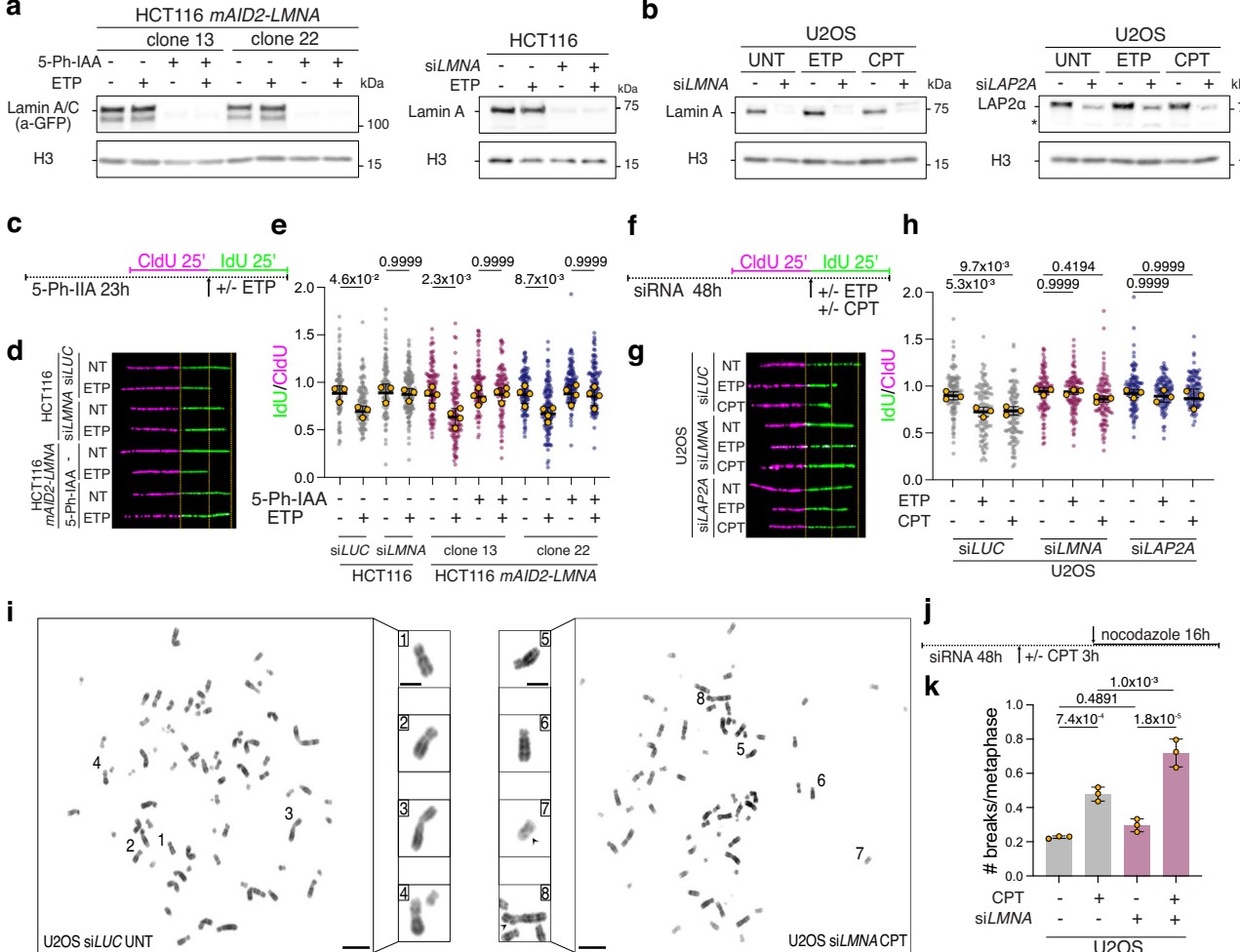

**Fig. 2 | Acute inactivation of Lamin A/C or LAP2α abolishes fork slowing and affects chromosomal stability upon mild RS. a**, **b** Western Blot analysis of Lamin A levels upon siRNA-mediated or 5-Ph-IAA-mediated depletion in the indicated cell lines. H3 is used as loading control. **c–e** DNA fiber analysis of HCT116 and HCT116 *mAID2-mClover-LMNA* cells upon siRNA-mediated or 5-Ph-IAA-mediated depletion of Lamin A/C. **c.** Schematic CldU/IdU pulse-labeling protocol used to evaluate fork progression upon 20 nM ETP. 5-Ph-IAA was added 24 h before the assay. **d** Representative DNA fiber images for the experiment in (**c**). **e** IdU/CldU ratio is plotted for a minimum of 100 forks from each of 3 (siRNA) and 4 (5-Ph-IAA) independent experiments. Yellow circles indicate the median for each experiment, while the black bar indicates the mean of the median values +/- SD. Statistical analysis was applied on the median values, using one-way ANOVA test with Bonferroni's *post hoc* correction. **f–h** DNA fiber analysis of U2OS cells upon siRNA-mediated depletion of Lamin A/C or LAP2α. **f** Schematic CldU/IdU pulse-labeling protocol used to evaluate fork progression upon 20 nM ETP or 100 nM CPT. siRNA

was transfected 48 h before the assay. **g** Representative DNA fiber images for the experiment in (**f**). **h** IdU/CldU ratio is plotted for a minimum of 100 forks from each of three independent experiments. Yellow circles indicate the median for each experiment, while the black bar indicates the mean of the median values +/- SD. Statistical analysis was applied on the median values, using one-way ANOVA test with Bonferroni's *post hoc* correction. **i** Representative metaphase spread images. Insets show magnified chromosomes, numbered in the overview images. Arrowheads point to chromosomal breaks. Scale bar: 10 μm, in inset 5 μm. **j** Schematic design of the metaphase spread experiment in (**i–k**). **k** Average number of chromosomal breaks in mock- or Lamin A/C-depleted (siRNA) U2OS cells, optionally treated with 100 nM CPT for 3 h followed by nocodazole treatment. Bar graph depicts mean +/- SD from 3 independent experiments (yellow dots). A minimum of 55 metaphases was analyzed per sample and experiment. Statistical analysis was applied on the median values, using one-way ANOVA test with Bonferroni's *post hoc* correction.

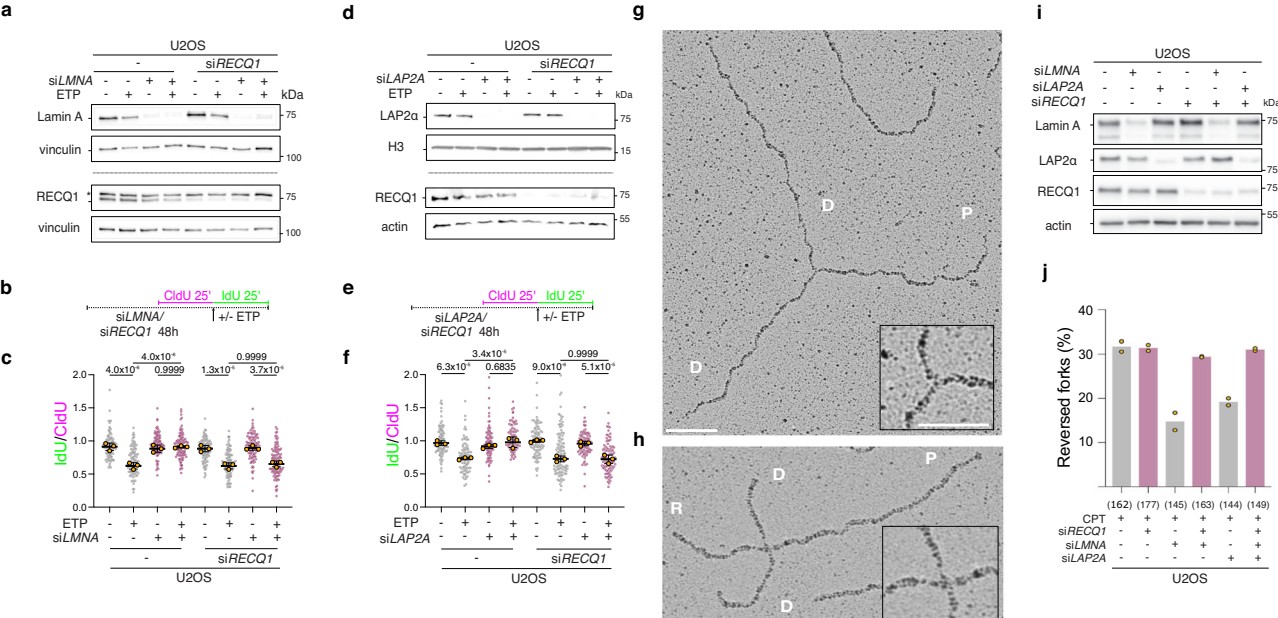

**Fig. 3 | Lamin A/C and LAP2α mediate fork slowing and reversal by limiting RECQ1-mediated fork restart. a–c** DNA fiber analysis of U2OS cells upon siRNA-mediated downregulation of *LMNA*, in combination with *RECQ1*. **a** Western Blot analysis of Lamin A and RECQ1 levels levels upon siRNA-mediated depletion for the experiment in a-b. Vinculin is used as loading control. *identifies an unspecific band occasionally recognized by the RECQ1 antibody. Western Blots were performed alongside each DNA fiber replicate (*n* = 3). **b** Schematic CldU/IdU pulse-labeling protocol used to evaluate fork progression upon 20 nM ETP. siRNAs were added 48 h before the assay. **c** IdU/CldU ratio is plotted for a minimum of 100 forks from each of 3 independent experiments. Yellow circles indicate the median for each experiment, while the black bar indicates the mean of the median values +/- SD. Statistical analysis was applied on the median values, using one-way ANOVA test with Bonferroni's *post hoc* correction. **d–f** DNA fiber analysis of U2OS cells upon siRNA-mediated downregulation of *LAP2A*, in combination with *RECQ1*. **d** Western Blot analysis of LAP2α and RECQ1 levels upon siRNA-mediated depletion for the experiment in d-e. H3 and actin are used as loading controls. Western Blots were performed alongside each DNA fiber replicate (*n* = 3). **e** Schematic CldU/IdU pulse-labeling protocol used to evaluate fork progression upon 20 nM ETP. siRNAs were added 48 h before the assay. **f** IdU/CldU ratio is plotted for a minimum of 100 forks from each of 3 independent experiments. Yellow circles indicate the median for each experiment, while the black bar indicates the mean of the median values +/- SD. Statistical analysis was applied on the median values, using one-way ANOVA test with Bonferroni's *post hoc* correction. **g, h** Electron micrographs of a representative normal replication fork (**g**) and reversed fork (**h**) from CPT-treated U2OS cells: parental (P) and daughter (D) duplexes, regressed arm (R). The insets show a magnification of the junction. Scale bars: in g 100 nm (inset 50 nm), in h 100 nm (inset 25 nm). **i** Western Blot analysis of Lamin A, LAP2α and RECQ1 levels upon siRNA-mediated depletion for the experiment in j. Actin is used as loading control. **j** Average frequency of reversed replication forks isolated from U2OS cells upon siRNA-mediated depletion of the indicated factors, and optional treatment with 100 nM CPT for 1 h. Yellow dots represent the observed percentage of reversed forks in each independent experiment (*n* = 2; see Supplementary Fig. 3g). Total number of molecules analyzed per condition in brackets.

instability, we used chromosome spreads from metaphase arrested cells and monitored chromosomal breaks and abnormalities (Fig. 2i). Using mild CPT treatments that induce per se mild chromosomal instability in U2OS cells, we observed a significant increase in chromosomal breakage upon *LMNA* downregulation (Fig. 2j, k). Overall, these data suggest that Lamin A/C and its nucleoplasmic interaction partner LAP2α are required to induce active fork slowing upon mild RS and to limit the associated genomic instability.

**Unrestrained fork progression in Lamin A/C-LAP2α depleted cells reflects deregulated fork restart by RECQ1**

Accelerated fork progression and/or defective fork slowing upon genotoxic treatments have been frequently reported to depend on uncontrolled Primpol activity, which rapidly re-primes DNA synthesis on extended ssDNA stretches and thereby prevents efficient replication fork reversal[19,39–41]. Hence, we tested by DNA fiber assays whether defective fork slowing upon Lamin A/C- or LAP2α inactivation may also reflect a similar deregulation. Surprisingly, effective *PRIMPOL* downregulation by siRNA in ETP-treated U2OS cells did not prevent the unrestrained fork progression induced by Lamin A/C- or LAP2α inactivation (Supplementary Fig. 3a–f), suggesting that defective fork slowing in this context does not reflect an altered equilibrium between fork reversal and repriming. An alternative mechanism reported to drive unrestrained fork progression upon stress depends on

deregulated restart of reversed forks by the RECQ1 helicase[6,7]. We tested RECQ1 contribution by co-downregulating it with *LMNA* or *LAP2A* in U2OS cells and found by DNA fiber assays that RECQ1 depletion fully rescued the unrestrained fork progression induced by *LMNA* or *LAP2A* downregulation, restoring active fork slowing in response to ETP treatment (Fig. 3a–f). These data suggest that reversed forks are efficiently formed upon genotoxic treatments but are untimely restarted by the deregulated action of RECQ1, when Lamin A/C or LAP2α are not functional. To test this hypothesis, we exploited an established single-molecule approach to directly visualize replication intermediates by electron microscopy[42,43], which allows distinguishing standard 3-way replication forks (Fig. 3g) from 4-way reversed forks (Fig. 3h). In line with previous results[6], fork reversal is detected at high levels upon CPT treatments (ca. 30% of the forks). Strikingly, reversed fork frequency drops to 10-20% when Lamin A/C or LAP2α are depleted, but is restored to control levels upon co-inactivation of RECQ1 (Fig. 3i, j; and Supplementary Fig. 3g). Collectively, these data strongly suggest that Lamin A/C and LAP2α are required in response to mild RS to actively slow down replication forks and maintain high levels of reversed forks, by negatively regulating RECQ1 fork restart activity. Indirectly, these data also show that reversed fork formation is perfectly functional also in the absence of Lamin A/C and LAP2α. Consistently, in our experimental conditions, Lamin A/C or LAP2α inactivation did not significantly affect the number or size of ssDNA

stretches detectable by EM at forks or behind them (ssDNA gaps), nor the chromatin loading of key ssDNA binding proteins (RPA/RAD51), which were previously reported as key intermediates or factors implicated in reversed fork formation (Supplementary Fig. 3h, i)[6].

## Lamin A/C-LAP2α limit fork progression upon stress promoting PARylation events at replication forks

RECQ1 was previously shown to be negatively regulated by its transient interaction with auto-modified (i.e., poly-ADP-ribosylated or PARylated) PARP1, thereby preventing the restart of reversed forks until the resolution of local replication stress[6,7]. Both Lamin A/C and LAP2α were recently identified by mass spectrometry as PARP1 proximal interactors and as proteins enriched at forks upon prolonged stalling[33]. We thus considered the hypothesis that Lamin A/C may globally regulate PARylation levels, thereby indirectly controlling RECQ1 activity upon replication stress. We confirmed that Lamin A/C and PARP1 physically interact via immunoprecipitation, although this interaction was not detectably altered in response to various treatments interfering with fork progression (Supplementary Fig. 4a). We then used an established procedure to preserve and detect the levels of protein PARylation in U2OS cell extracts, which is expectedly affected by cellular treatments with PARP- and PARG-inhibitors (Supplementary Fig. 4b). Neither mild ETP treatment nor *LMNA* downregulation in U2OS cells - in the same experimental conditions that strongly impacted on replication fork progression (Figs. 2, 3) - significantly altered global protein PAR/MARylation levels in this assay, as detected by two different antibodies (96-10, E6F6A; Supplementary Fig. 4b), suggesting that the control of replication fork progression does not entail major changes in global PARylation events. Stabilization and detection of protein PAR/MARylation is sensitive to specific procedural steps and detection reagents[44]. Hence, we also used a different established procedure to isolate ADP-ribosylated proteins and detect them via an engineered ADP-ribose binder[45]. In these experimental conditions, we reproducibly observed a significant reduction in protein PARylation upon *LMNA* downregulation, while pre-treatment with a specific PARG inhibitor (PARGi) restored physiological PAR levels in Lamin A/C depleted cells (Supplementary Fig. 4c, d).

These data suggested to us that a specific subset of PARylation events – which may be under/over-represented depending on the experimental procedure – could be modulated by Lamin A/C and possibly responsible for the control of fork progression in response to mild RS. To test this hypothesis, we specifically detected PAR in proximity to replication factories as previously described[34], i.e. via PLA assays detecting the proximity of PAR (E6F6A antibody) to nascent DNA (EdU). In this assay, mild CPT treatment did not significantly increase PAR levels in proximity to EdU, but *LMNA* downregulation markedly decreased the number and the intensity of PLA signal in both treated and untreated cells, showing a similar effect to the treatment with the PARP inhibitor Olaparib (Fig. 4a–c; Supplementary Fig. 4e, f). Treatment with PARGi, as expected, increased the detectable level of PAR in proximity to replication forks, but also fully suppressed its reduction induced by *LMNA* downregulation (Fig. 4d–f). Hence, to assess whether the Lamin A/C-dependent control of PAR levels at replication forks is functionally relevant to modulating replication fork progression upon stress, we used PARGi treatment in DNA fiber assays upon mild ETP treatments. In contrast to first generation PARG inhibitors[46], treatment with the specific PARGi used in this study did not alter fork progression per se, but was sufficient to fully restore active fork slowing in ETP-treated Lamin A/C-depleted cells (Fig. 4g–i). Moreover, PARGi treatment also fully suppressed the unrestrained fork progression induced by *LAP2A* downregulation upon ETP treatment (Supplementary Fig. 4g, h). Collectively, these results strongly-suggest that Lamin A/C and LAP2α control a specific subset of PARylation events at replication forks, which is functionally relevant to control RECQ1 activity and mediate active fork slowing upon stress.

## Defective fork heterochromatinization upon genotoxic stress phenocopies Lamin A/C/LAP2α inactivation

Along with histone modifications, Lamin A/C and LAP2α were shown to alter chromatin dynamics and diffusion[28,47], which are emerging as key modulators of DNA repair and DNA replication mechanisms[48]. Transient compaction of replicating chromatin via deposition of heterochromatic marks (e.g., H3K9me3) was recently shown to assist the cellular response to prolonged fork stalling[13]. Moreover, PARP1 efficiently binds heterochromatin at specific genomic domains and gets activated within condensed chromatin to promote DNA repair[49,50]. Based on these findings, we set out to investigate i) whether accumulation of heterochromatin marks on nascent DNA may also occur in RS conditions that are permissive for residual fork progression; ii) whether chromatin modifications could be implicated in this novel role of nucleoplasmic Lamin A/C regulating fork restart upon mild RS.

To address the first point experimentally, we tested in HCT116 cells whether G9a/EHMT2–i.e., the lysine methyl transferase mediating the first steps of H3K9 methylation[51] – is recruited to replicating DNA upon mild RS. EdU-PLA experiments showed that low concentration treatments with CPT or ETP do induce significant recruitment of G9a to nascent DNA, comparable to the one reported upon HU-induced fork stalling (Fig. 5a, b)[13]. Accordingly, PLA experiments upon the same treatments showed a significant accumulation on nascent DNA of the heterochromatic marker H3K9me3 (Fig. 5c, d), which is mediated by the sequential action of G9a and SUV39H1 methyltransferases[13]. These data show that accumulation of heterochromatin marks on newly replicated DNA does not require fork stalling and is also observed upon mild conditions of RS, i.e. those we used here to uncover the role of Lamin A/C in modulating fork remodeling and restart. To test the possible involvement of chromatin compaction in these mechanisms, we performed DNA fiber spreading experiments using UNC0642, a specific and potent catalytic inhibitor of G9a[52] (G9ai), which was previously used to establish the functional role of chromatin compaction upon fork stalling[13]. Remarkably, we found that – analogously to Lamin A/C and LAP2α inactivation (Figs. 2–4) – G9a inhibition leads to unrestrained fork progression upon mild CPT and ETP treatments, and that this defect is suppressed by PARG inhibition (Fig. 5e, f). Similarly to Lamin A/C inactivation, PAR levels at replication factories are reduced by G9a inhibition in CPT-treated U2OS cells and restored by simultaneous inhibition of PARG (Fig. 5g). Moreover, the unrestrained fork progression observed upon G9a inhibition depends on RECQ1 (Fig. 5h, i). Finally, as observed upon Lamin A/C-LAP2α inactivation, G9a inhibition markedly affected the accumulation of CPT-induced reversed forks, which was restored by concomitant PARG inhibition. Conversely, G9a inhibition did not affect reversed fork frequency upon short (1 h) HU treatments that do not trigger reversed fork degradation, but that prevent RECQ1-dependent restart via nucleotide depletion (Fig. 5j and Supplementary Fig. 5a). Altogether, these data consolidate a striking phenocopy of G9a and Lamin A/C-LAP2α inactivation for the control of local ADP ribosylation and replication fork restart (see also Figs. 2–4).

## Lamin A/C promotes heterochromatin maintenance at forks, thereby modulating RECQ1 via ADP-ribosylation

Based on these data and on the established role of Lamin A/C and LAP2α in modulating chromatin dynamics within the nucleoplasm[28,47], we hypothesized that nucleoplasmic Lamin A/C may modulate fork restart upon mild RS by mediating chromatin compaction at replication forks. H3K9me3-PLA experiments confirmed that Lamin A/C depletion in our *HCT116 mAID2-mClover-LMNA* cell line markedly reduces the levels of H3K9me3 detected at forks upon CPT treatment (Fig. 6a, b).

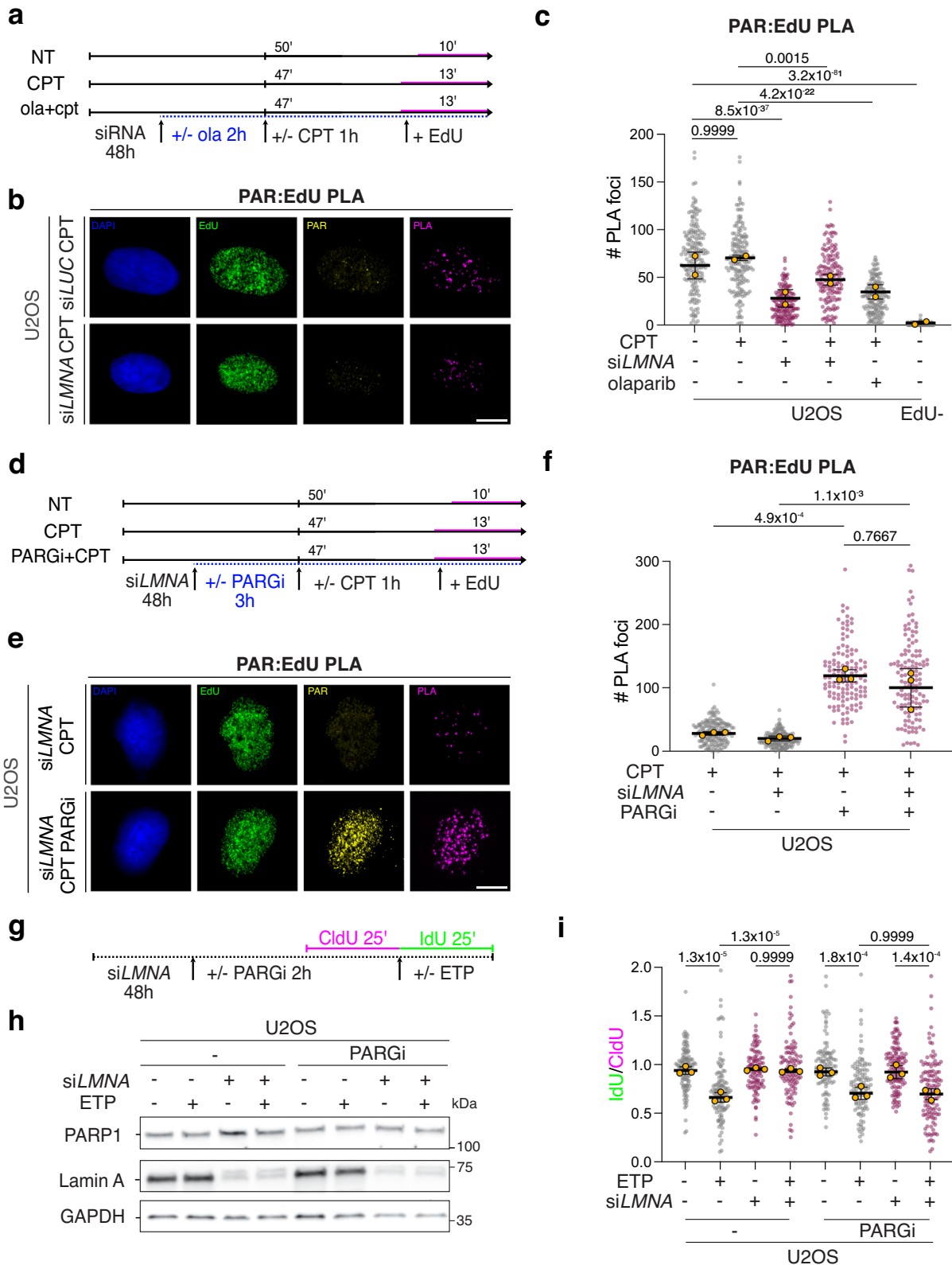

Moreover, using *ChromStretch*–a single molecule method to map proteins and epigenetic marks directly on individual replication tracks[13]–we confirmed both in U2OS and in *HCT116 mAID2-mClover-LMNA* cells that mild CPT treatment leads to a marked accumulation of H3K9me3 at replication forks, and that Lamin A/C inactivation drastically impairs the levels of H3K9me3 accumulation (Fig. 6c–e and Supplementary Fig. 5b–f). Relative chromatin density typically decreases during replication, reflecting local chromatin opening as forks progress. However, even under mild replication stress (25 nM CPT), increased H3 density and H3K9me3 deposition at a significant fraction of the tracks suggest compaction and stabilization of nascent chromatin into a heterochromatic state. Lamin A/C depletion disrupts this process, resulting in reduced histone density and loss of H3K9me3, indicating impaired de novo heterochromatin formation at stressed replication forks.

**Fig. 4 | Lamin A/C sustains PAR levels at replication forks, thereby allowing fork slowing upon mild RS. a** Experimental design for the IF/PLA experiment in (**b**, **c**). In a and d, the duration of the EdU pulse is adapted to allow comparable incorporation of EdU despite the genotoxic treatments. The PARP inhibitor olaparib (ola; 10 μM) is used as positive control of reduced PAR accumulation on nascent DNA. **b** Representative U2OS nuclei (DAPI) upon optional *LMNA* downregulation, treated for 1 h with 100 nM CPT and stained for DNA synthesis (EdU), PAR and its physical proximity to nascent DNA (PAR:EdU PLA). Scale bar: 10 μm. **c** Quantification of PAR:EdU PLA signals from a-b. Signal was quantified in at least 100 EdU+ nuclei, in each of the 3 independent experiments. Yellow circles indicate the median for each experiment, while the black bar indicates the mean of the median values +/- SD. Statistical analysis was applied on the median values, using one-way ANOVA test with Bonferroni's *post hoc* correction. EdU- cells are used as negative control. **d** Experimental design for the IF/PLA experiment in (**e**, **f**). **e** Representative U2OS nuclei (DAPI) after *LMNA* downregulation, treated for 1 h with 100 nM CPT and optionally with the PARG inhibitor (PDD0017272, 1 μM),

stained as in (**b**). **f** Quantification of PAR:EdU PLA signals from d-e. Signal was qunatified in at least 100 EdU+ nuclei, in each of the 3 independent experiments. Yellow circles indicate the median for each experiment, while the black bar indicates the mean of the median values +/- SD. Statistical analysis was applied on the individual experiments, using Kruskal-Wallis test with Dunn's *post hoc* correction. EdU- cells are used as negative control. Scale bar: 10 μm. **g–i** DNA fiber analysis of U2OS cells upon siRNA-mediated downregulation of *LMNA*, and optional treatment with the PARG inhibitor (PDD0017272, 1 μM). (**g**). Schematic CldU/IdU pulse-labeling protocol used to evaluate fork progression upon 20 nM ETP. siRNA was added 48 h before the assay, while PARGi was added 2 h before. **h** Western Blot analysis of Lamin A levels upon siRNA-mediated depletion. Actin is used as loading control. PARP1 levels are not affected by PARGi treatment. **i** IdU/CldU ratio is plotted for a minimum of 100 forks from each of 3 independent experiments. Yellow circles indicate the median for each experiment, while the black bar indicates the mean of the median values +/- SD. Statistical analysis was applied on the median values, using one-way ANOVA test with Bonferroni's *post hoc* correction.

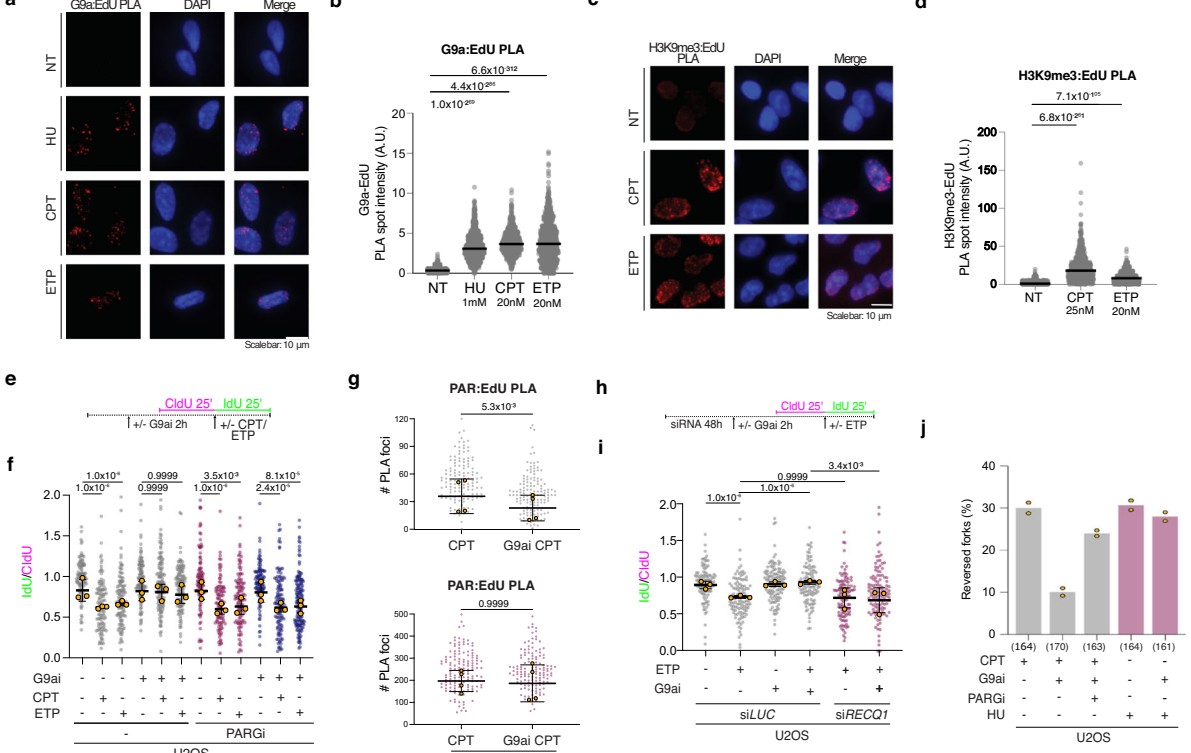

**Fig. 5 | G9a-dependent H3K9me3 accumulates on nascent DNA upon mild RS and limit RECQ1 fork restart activity via ADP ribosylation. a** Representative PLA images illustrating G9a enrichment on nascent DNA upon mild RS (G9a-EdU PLA, red). RPE-1 cells were labeled with EdU for 20 min either before optional treatment with 1 mM HU (1 h) or at the end of 20 nM CPT (1 h) and 20 nM ETP (1 h) treatment. Scale bar: 10 μm. **b** Quantification of the total intensity of all G9a-EdU PLA spots per nucleus in (**a**). In (**b**, **d**), *n* > 800 S phase cells were analyzed in each condition; Kruskal- Wallis test followed by Dunn's test were performed to test statistical significance for each of 2 independent PLA experiments. **c** Representative PLA images illustrating H3K9me3 deposition on nascent DNA upon mild RS (H3K9me3-EdU PLA, red). *mAID2-LMNA* HCT116 cells were labeled with EdU for 20 min at the end of optional treatment with 25 nM CPT (1 h) and 20 nM ETP (1 h). Scale bar: 10 μm. **d** Distribution of H3K9me3-EdU total PLA spot intensity per nucleus. Statistical analysis as in (**b**). **e** Schematic CldU/IdU pulse-labeling protocol used in f to evaluate fork progression upon treatment with 20 nM ETP, 100 nM CPT, G9ai (UNC0642, 1 μM), and/or PARGi (PDD0017272, 1 μM). G9ai and PARGi were added 2 h before the assay. **f** IdU/CldU ratio is plotted for a minimum of 100 forks from each of 3 independent experiments. Similar results were observed in all independent experiments. Yellow circles indicate the median for each experiment, while the black bar indicates the mean of the median values +/- SD. Statistical analysis was applied on the median values, using one-way ANOVA test with Bonferroni's *post hoc*

correction. **g** Quantification of PAR:EdU PLA signals from U2OS cells, treated with 100 nM CPT, G9ai (UNC0642, 1 μM) and PARGi (PDD0017272, 1 μM). Experimental design as in Fig. 4a, d. Signal was quantified in at least 100 EdU+ nuclei, in each of the 3 independent experiments. PARGi-treated and untreated samples were processed in parallel, but are displayed in different graphs due to different ranges of observed signal. Yellow circles indicate the median for each experiment, while the black bar indicates the mean of the median values +/- SD. Statistical analysis was applied on the individual experiments, using Kruskal-Wallis test with Dunn's *post hoc* correction. **h** Schematic CldU/IdU pulse-labeling protocol used in i to evaluate fork progression upon treatment with 20 nM ETP and G9ai (UNC0642, 1 μM), and/ or RECQ1 downregulation by siRNA. siRNA was transfected 48 h before the assay, while G9ai was added 2 h before. **i** IdU/CldU ratio is plotted for a minimum of 100 forks from each of 3 independent experiments. Yellow circles indicate the median for each experiment, while the black bar indicates the mean of the median values +/- SD. Statistical analysis was applied on the median values, using one-way ANOVA test with Bonferroni's *post hoc* correction. **j** Average frequency of reversed replication forks isolated from U2OS cells treated for 1 h with 100 nM CPT or 2 mM HU, combined with the indicated inhibitors. Yellow dots represent the observed percentage of reversed forks in each independent experiment (*n* = 2; see Supplementary Fig. 5a). Total number of molecules analyzed per condition in brackets. *A.U.*: arbitrary units.

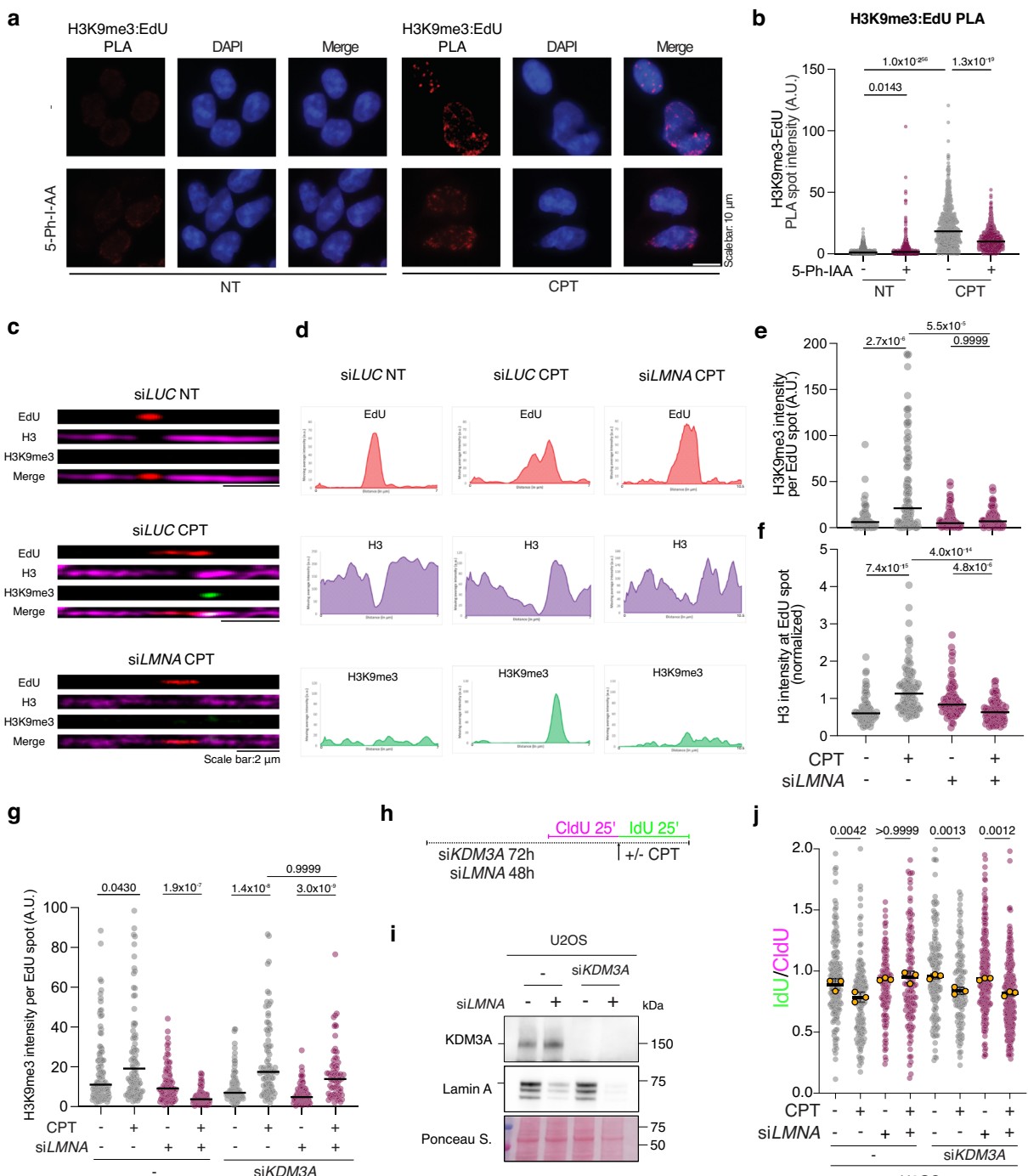

**Fig. 6 | Lamin A/C is required to accumulate H3K9me3 at forks upon mild RS, via modulation of the KDM3A demethylase. a** Representative PLA images illustrating H3K9me3 deposition on nascent DNA upon mild RS (H3K9me3-EdU PLA, red) in *mAID2-LMNA* HCT116 cells. Cells were treated with 5-Ph-IAA for 24 h prior to the experiment, to induce *Lamin A/C* depletion. Cells were pulsed with EdU for 20 min at the end of the optional treatment with 25 nM CPT (1 h). Scale bar: 10 μm. **b** Quantification of H3K9me3-EdU total PLA spot intensity per nucleus. *n* > 800 S phase cells were analyzed in each condition; Kruskal–Wallis test followed by Dunn's test were performed to test statistical significance for each of 2 independent PLA experiments. **c** Representative images of chromatin fibers acquired by Chrom-Stretch stained for EdU (red), H3 (magenta) and H3K9me3 (green), from U2OS cells: untreated (top), treated with 25 nM CPT (1 h) (middle), and upon downregulation of *LMNA* prior to treatment with 25 nM CPT (1 h) (bottom). Cells were pulsed with EdU for 20 min at the end of the optional treatments with 25 nM CPT (1 h). Scale bar: 2 μm. **d** Moving average intensity profiles of EdU (red), H3K9me3 (green) and H3 (magenta) of the representative fibers shown in (**c**). **e** Quantification of H3K9me3 signal overlapping with EdU spots. *n* > 75 EdU tracks for each condition were

analyzed in 3 independent experiments. **f** Quantification of H3 signal overlapping with EdU spots (normalized to the H3 signal outside EdU bubble). *n* > 75 EdU tracks for each condition were analyzed in 3 independent experiments. **g** Quantification of H3K9me3 signal overlapping with EdU spots upon optional treatment with 25 nM CPT (1 h) and optional downregulation of *LMNA* or *KDM3A*. *n* > 70 EdU tracks for each condition were analyzed in two independent experiments. Kruskal-Wallis test followed by Dunn's test were performed to test statistical significance for PLA and chromatin fiber analysis. **h** Schematic CldU/IdU pulse-labeling protocol used in j to evaluate fork progression upon treatment with 100 nM CPT and/or *KDM3A* downregulation by siRNA. **i** Western Blot analysis of Lamin A and KDM3A levels upon siRNA-mediated depletion for the experiment in j. Ponceau S. is shown as loading control. **j** IdU/CldU ratio is plotted for a minimum of 100 forks from each of 3 independent experiments. Yellow circles indicate the median for each experiment, while the black bar indicates the mean of the median values +/- SD. Statistical analysis was applied on the median values, using one-way ANOVA test with Bonferroni's *post hoc* correction. *A.U.* arbitrary units.

(Fig. 6c–f and Supplementary Fig. 5b–g). Interestingly, we performed similar experiments in U2OS cells upon downregulation of Jumonji domain-containing protein 1 A (JMJD1A)/Lysine (K)-Specific Demethylase 3 A (KDM3A)–i.e. the demethylase shown to remove H3K9me3 from stalled forks during restart[13]–and found that impairing H3K9me3 demethylation was sufficient to fully suppress the defect in H3K9me3 levels at forks induced by Lamin A/C inactivation (Fig. 6g–i). Consistently, KDM3 downregulation in U2OS cells restores CPT-induced active fork slowing in lamin A/C defective cells (Fig. 6j). These results suggest that accumulation of heterochromatin marks at forks is required to modulate fork progression and restart upon mild RS and is modulated by Lamin A/C, likely at the level of H3K9 demethylation, impacting the maintenance of the epigenetic mark on replicated DNA.

## Discussion

Our data establish a new important role for Lamin A/C in supporting active fork slowing and genome stability upon mild replication interference, by modulating RECQ1 activity at reversed forks. Defective replication fork reversal was not reported in Lamin A/C-depleted cells upon fork stalling by HU treatment[31]. It should be noted, however, that RECQ1 activity is very limited in these conditions, as nucleotide depletion impairs efficient fork restart[6]. The phenocopy observed for Lamin A/C and LAP2α inactivation–along with the pan-nuclear interaction with replication factories and the global effects on replication fork progression–strongly suggest that this novel role of Lamin A/C in the RS response entails primarily the nucleoplasmic fraction of the protein. Due to its intrinsic solubility and heterogeneity in structure[53], nucleoplasmic Lamin A/C is much harder to detect by conventional imaging methods; based on our findings, visualizing at high resolution the organization of Lamin A/C structures at replication factories will represent an important and exciting challenge for future studies. It is in principle surprising that this function of Lamin A/C in response to mild RS correlates with decreased proximity of the protein to short stretches of nascent DNA, as those induced in the experimental conditions of our PLA assays (Fig. 1). We propose that the observed reduction in Lamin A/C-EdU PLA signal upon mild RS does not imply release of the protein from replication factories, but instead reflects a different spatial arrangement of the protein in respect to nascent DNA (Fig. 7). This distance may be transiently increased by replication fork remodeling, as elegantly shown for the replicative helicase in similar PLA assays[36]. Although we expect these Lamin A/C-mediated events to take place throughout the nucleoplasm, our data do not exclude that the protein could exert a similar role also within the lamina, possibly assisting specific fork restart mechanisms that were shown to entail fork relocation to the nuclear membrane[54].

Our data also highlight the functional role of PARylation events in the control of replication fork progression and restart, suggesting that local PAR accumulation within replication factories is required to control RECQ1 activity and thereby mediate active fork slowing throughout the nucleus. This evidence extends previous studies implicating PAR-mediated mechanisms in replication fork progression, remodeling and restart, even in response to perturbations that do not prevent bulk DNA synthesis, such as defective Okazaki fragment ligation[6,7,34,35,55]. How could Lamin A/C control PARylation levels within replication factories? Lamin A/C defects were shown to induce altered levels of NAD +, a crucial co-substrate for PAR synthesis, but this function is tissue-specific and related to mitochondrial defects[56,57]. Two lines of evidence in our work rather suggest that Lamin A controls PARylation at replication forks via modulation of local chromatin compaction and accessibility: i) defective H3K9 methylation affects PAR levels at forks even in Lamin A/C-proficient cells and strikingly phenocopies Lamin A/C inactivation, ii) replication defects upon both types of genetic perturbation reflect the deregulated action of RECQ1 and can be suppressed by PARG inhibition, which restores high PAR levels at replication factories. PARP1 is very efficient at PARylating itself

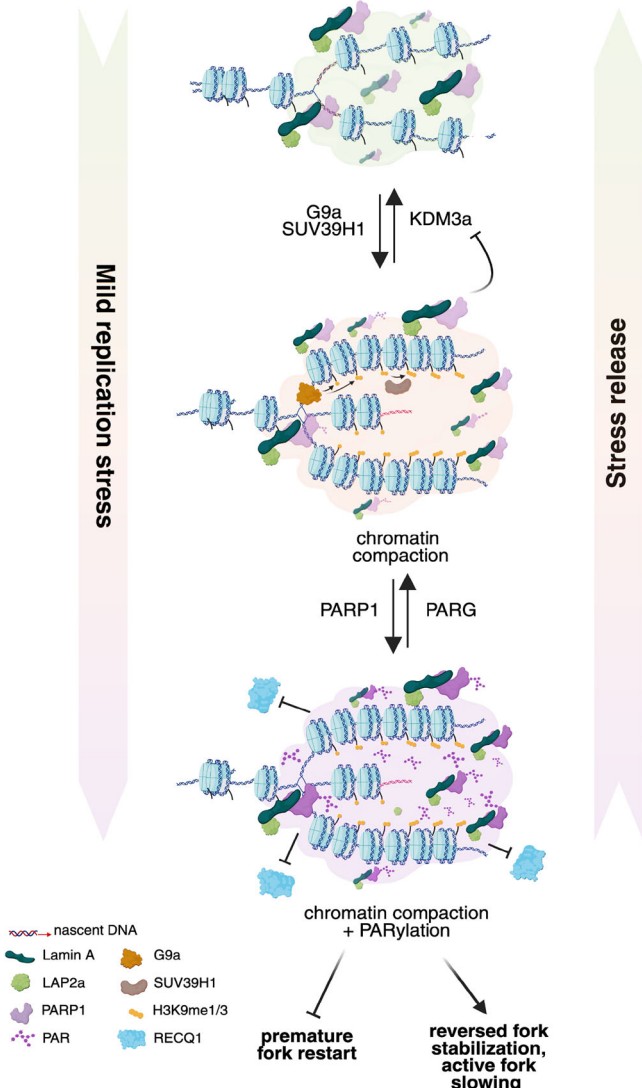

**Fig. 7 | Model for the role of Lamin A/C and LAP2α limiting RECQ1-mediated fork restart upon mild RS, by increasing H3K9me3 and ADP ribosylation levels at replication factories.** Under normal conditions, nucleoplasmic Lamin A/C and its partner LAP2α dynamically associate with replication factories, maintaining replication competence across the nucleus. Upon mild replication stress, Lamin A/C becomes essential for enforcing replication fork slowing by promoting the accumulation of the heterochromatin mark H3K9me3 -mediated by G9ai/SUV39H1 H3K9 methylases- and supporting local PARylation at nascent DNA. Lamin A/C prevents premature removal of H3K9me3 by inhibiting the demethylase JMJD1A/KDM3A, while H3K9me3 in turn facilitates PARP activation. PARylation serves to limit the engagement of the RECQ1 helicase at replication forks, thereby stabilizing reversed forks, promoting active fork slowing and suppressing untimely RECQ1-mediated fork restart. Both H3K9me3 accumulation and ADP ribosylation are reversible processes, that can be reverted respectively by KDM3A and PARG once the stress is released. Loss of Lamin A/C, H3K9me3, or PARylation disrupts this coordinated response, leading to unregulated fork restart and genomic instability.

and this modification is sufficient to inhibit RECQ1 enzymatic activity in vitro[7], representing a critical regulatory event for fork restart. However, the concomitant accumulation of PAR and histone methylation events at replication forks upon mild RS suggests that additional PARylation events on unknown protein targets or DNA could mediate RECQ1 inhibition within active replication centers, transiently preventing its efficient engagement at replication forks and thereby stabilizing reversed forks to promote fork slowing. Uncovering the molecular mechanisms achieving complete control of RECQ1

recruitment and activity at replication forks will require further investigation. Considering that RECQ1 is the most abundant of the RECQ family helicases in human cells and a key factor mediating the balance between fork slowing and restart, it seems likely that its activity entails multiple levels of control.

Collectively, our data support a sequential model where upon mild RS Lamin A/C promotes efficient PARylation and RECQ1 control at replication factories by assisting chromatin methylation and compaction (Fig. 7). Lamin A/C was previously shown to modulate chromatin organization and gene expression by restricting chromatin mobility and diffusion[47]. Lamin A/C directly interacts with SUV39H1 and stabilizes it, thereby affecting H3K9me3 levels and chromatin compaction[29]. Moreover, Lamin A/C interacts with the histone deacetylases HDAC2 and SIRT6, modulating heterochromatinization and promoting PARylation, as well as DNA repair[58,59]. Despite this insight, the detailed mechanisms by which Lamin A/C regulates the deposition or maintenance of heterochromatin marks in different contexts are yet elusive. Here we provide evidence that, in the context of replication factories and mild RS, both decreased H3K9me3 levels and defective fork slowing in Lamin A/C-depleted cells can be rescued by inactivation of the KDM3a demethylase, suggesting that Lamin A/C controls chromatin compaction mainly by limiting KDM3a access or activity at replication factories, thereby stabilizing the heterochromatin mark on genomic regions that experience mild RS (Fig. 7). Upon acute inactivation of Lamin A/C, KDM3a-dependent H3K9 demethylation impairs H3K9me3 accumulation at forks experiencing RS, which in turn affects the equilibirium between PAR synthesis and degradation, reducing PAR levels at replication factories. This ultimately deregulates access and/or activity of RECQ1 at forks that had been reversed upon mild RS, leading to unrestrained fork progression and genomic instability (Supplementary Fig. 6). Although based on our evidence we graphically depict this as a linear sequence of events (Fig. 7), it is plausible that nucleoplasmic lamin A/C, heterochromatin marks and ADP ribosylation regulate each other through interdependent mechanisms that are fine-tuned by feedback loops. For instance, lamin A/C levels at replication factories were found to be reduced upon HU treatments under conditions of defective H3K9 methylation[13], suggesting that a positive feedback loop may contribute to the severe replication phenotypes observed upon chromatin decompaction and replication stress. It is also likely that these molecular events are differently modulated across cell types, depending on specific gene expression programs, basal chromatin accessibility, replication timing profiles, epigenetic landscape and ADP ribosylation dynamics.

Beyond the specific role of Lamin A/C in the modulation of replicating chromatin, our data establish a novel, general link between chromatin compaction and PARylation in response to mild RS. How RS-induced heterochromatic marks promote PARylation within replication factories and vice versa is currently unclear. PARP1 binding was previously reported at various specialized heterochromatic domains, such as telomeres, centromeres and silent rDNA repeats, promoting local and functionally relevant ADP-ribosylation events[49]. Moreover, excessive accumulation of heterochromatin at replication forks was shown to induce local ADP ribosylation by promoting discontinuous DNA synthesis[60,61], which is a potent activator of PARP1[34]. Intriguingly, PARP1 activation was recently shown to follow its initial condensation at damaged sites to promote the subsequent DNA repair steps[50], suggesting that PARP1 may be recruited and subsequently activated within condensed chromatin. Although our data suggests that chromatin compaction precedes and mediates sufficient levels of PARylation at replication forks, they do not exclude that PARP1 activation may consolidate a protected subnuclear environment within those factories, limiting access to abundant and potentially deleterious proteins, as proposed for Primpol upon fork stalling[13] and here for RECQ1 upon mild RS. Interestingly, PARP1 was shown to promote nucleosome

assembly in vitro[62] and to induce local chromatin compaction and silencing in cells[63]. Moreover, several chromatin remodelers promoting a compact chromatin environment are known as PARP1 targets[64,65], suggesting that efficient PARylation may further consolidate chromatin compaction in replicating domains facing RS. Based on our evidence, uncovering the complex mechanistic crosstalk of chromatin compaction, PARylation and replication fork progression will represent a promising and exciting avenue of future research.Recent findings uncovered novel functions for Lamin A/C in the ATR-mediated control of nuclear and micronuclear membrane rupture upon excessive DNA damage[66,67]. Our study uncovers another independent function of Lamin A/C, protecting genome stability throughout the nucleus in response to mild genotoxic treatments. The ATR kinase was shown to modulate fork progression and remodeling upon mild RS[20], but the underlying mechanisms have remained elusive. Importantly, upon HU-induced fork stalling, ATR was also shown to mediate chromatin compaction on replicated DNA[13], and to induce the relocation of stalled forks towards the nuclear periphery via nuclear F-actin polymerization[16]. Investigating how ATR signaling modulates nuclear architecture and organization throughout the nucleoplasm upon mild replication interference will likely reveal important regulatory mechanisms of the RS response. Nucleoplasmic Lamin A/C may act upstream or in coordination with ATR signaling, serving as a structural and regulatory scaffold that shapes the chromatin environment, enforcing fork slowing under mild replication stress via maintenance of H3K9me3 and promotion of local PARylation. This epigenetic and enzymatic environment prevents premature RECQ1-dependent fork restart, ensuring genome stability. Loss of any component −Lamin A/C, H3K9me3, or PAR− disrupts this network, leading to aberrant fork dynamics and genomic instability. Gaining molecular insight on the role of nucleoplasmic Lamin A/C upon genotoxic stress may provide novel mechanistic explanations for the dramatic consequences of Lamin A/C deregulation in laminopathies and aging, so far primarily attributed to defects in the structural support of the nuclear lamina. Importantly, further investigation of the Lamin A/C−H3K9me3−PARP1 axis may shed light on the molecular underpinnings of replication stress tolerance and uncover novel vulnerabilities in response to chemotherapeutics, especially PARP inhibitors, with potential implications for improving treatment strategies in cancer.

## Methods
### Key materials
**Antibodies.** anti-Histone H3, Abcam Cat # ab1791
anti-Actin, Sigma-Aldrich Cat # A5441
anti-Tubulin, Sigma-Aldrich Cat # T5168
anti-Vinculin, Thermo Fischer Scientific Cat # 700062
anti-RECQ1, Novus Biologicals Cat # NB100-618
Rat anti-Primpol, kindly provided by J. Méndez
Rat anti-BrdU (CldU), Abcam Cat # ab6326
anti-SMC1, Thermo Fischer Scientific Cat # PA5-29122
Mouse anti-BrdU (IdU), Becton Dickinson Cat # 347580
Donkey anti-rat-Cy3, LubioScience Cat # 712-166.153
anti-GFP, Abcam # ab290
Mouse anti-gH2AX (Ser139), Millipore Cat # 05-636
anti-Lamin A, SantaCruz Biotechnology Cat #L12923
anti-Lamin A/C, Proteintech, Cat # 10298-1-AP
anti-Lamin A/C (E-1), SantaCruz Biotechnology Cat # SC-376248
anti-Lamin A, Abcam Cat # ab83472
anti-PARP1, Tulip Biolabs, Cat #2090
anti-PARP1 (C2-10), produced in-house, kindly provided by J.-P. Gagné and G. G. Poirier
anti-LAP2α, Abcam Cat # ab5162
anti-PAR CST (E6F6A), Cell Signaling Technology Cat # 83732
anti-PAR 96-10, produced in-house, kindly provided by J.-P. Gagné and G. G. Poirier

anti-MAR/PAR eAf1521-Fc fusion protein, kindly provided by M. O. Hottiger
anti-PARP1 (9571), kindly provided by J.-P. Gagné and G. G. Poirier
anti-H3K9me2, Active Motif Cat # 39754
anti-H3K9me3, Abcam Cat # ab176916
anti-G9a (EPR18894), Abcam Cat # ab185050
anti-IgG mouse, SantaCruz Biotechnology Cat # sc-2025
anti-KDM3A/JMJD1A, Proteintech, Cat # 12835-1-AP
anti-RPA32/RPA2, Abcam Cat # ab2175
anti-RAD51, BioAcademia Cat # 70-001
ECL anti-Rabbit IgG, Horseradish Peroxidase linked whole antibody, Amersham Cat # NA934V
ECL anti-Mouse IgG, Horseradish Peroxidase linked whole antibody, Amersham Cat # NA931V
Goat anti-mouse- Alexa Fluor™ 488, Thermo Fisher Scientific Cat # A-11001
Goat anti-rabbit- Alexa Fluor™ 488, Thermo Fisher Scientific Cat # A-11001
Goat anti-mouse Alexa Fluor™ 555, Azide Thermo Fisher Scientific Cat # A-21422
Goat anti-rabbit Alexa Fluor™ 555, Azide Thermo Fisher Scientific Cat # A-21428

**Chemicals.** RNAiMAX, Thermo Fisher Scientific Cat # 13778075
Camptothecin, Sigma-Aldrich Cat # C9911
Etoposide, Sigma-Aldrich Cat # E1383
5-Phenyl-1H-indole-3-acetic acid (5-Ph-IAA), Bioacademia Cat # 30-003
Benzyldimethylalkyl Ammonium Chloride, Sigma-Aldrich Cat # B6295
Formamide, Sigma-Aldrich, Cat # 47680
Glutaraldehyde 25%, EMS Cat # 16200
Uranyl acetate, Fluka Cat # 73943
5-Chloro-2′-deoxyuridine, Sigma-Aldrich Cat # C6891
5-Iodo-2′-deoxyuridine, Sigma-Aldrich Cat # I7125
Nocodazole, Sigma-Aldrich Cat # M1404
ProLong Gold Antifade Mountant Thermo Fisher Scientific Cat # P36930
ibidi mounting medium, ibidi, # Cat 50001
Western Bright ECL-HRP Substrate, Advansta Cat # K-12045
WesternBright Sirius - femtogram HRP Substrate, Advansta, Cat # K-12043
DAPI, Sigma-Aldrich Cat # D9542
Nonidet™ P-40, Merck, Cat # 21-3277
Dynabeads™ Protein G for Immunoprecipitation, Thermo Fischer Scientific Cat #10003D
Benzonase, Millipore, Cat # E1014
Biotin-azide, Merck # Cat 762024
Biotin-azide, Jackson ImmunoResearch Cat # AB_2339006
G9ai-UNC0642, MedChemExpress, kindly provided by Taneja lab
PARGi PDD0017272, Lucerna-Chem, Cat # HY-133531
PARPi-olaparib, Selleckchem Cat # S1060
Duolink In Situ PLA Probe Anti-Rabbit PLUS, Merck Cat # DUO92002
Duolink In Situ PLA Probe Anti-Mouse MINUS, Merck Cat # DUO92004
Duolink In Situ Detection Reagents Red, Merck Cat # DUO92008
Poly-L-lysine, Sigma-Aldrich Cat # P4832.

## Cell lines
Human osteosarcoma U2OS cells and human colon cancer HCT116 cells were acquired from ATCC. The HCT116 F74G cell line was kindly provided by the Kanemaki laboratory. The mAID2-mClover-LMNA HCT116 F74G cell line was generated in Massimo Lopes' laboratory.

## Cell culturing
U2OS cells and HCT116 cells were cultured in Dulbecco's Modified Eagle Medium (DMEM, 41966-029, Life Technologies) supplemented with 10% Fetal Bovine Serum (FBS, GIBCO), 100 U/mL penicillin and 100 mg/mL streptomycin at 37 °C in a humidified atmosphere containing 6% CO2.

## Cell line generation
The mAID2-mClover-LMNA HCT116 F74G cell line was generated as described previously (Natsume et al., 2016, Yesbolatova et al., 2019, Yesbolatova et al., 2020). The gRNA was cloned into the CRISPR-Cas9 containing plasmid pX330 (Addgene Cat # 42230-DNA.cg) according to Ann et al. 2016, using the following oligos for the N-terminal region: 5′-CGCTGCCAACCTGCCGGCCA-3′ (CRISPR gRNA) and 5′-TGGCCGGCAGGTTGGCAGCG-3′ (reverse complement). For the donor plasmid, we used HCT116 genomic DNA to amplify by PCR and clone into the pJET plasmid (CloneJET PCR Cloning Kit, Thermo Fisher Scientific Cat # K1232) a 1 kb fragment as homology arms (HAs) at the N-terminus of the LMNA gene containing the first ATG codon, using the following oligos: 5′-CACCCACTCTCCCTCCTTGG-3′ (forward primer) and 5′-GCCCCAACTTGTCCCTGATAC-3′ (reverse primer). The HA containing plasmid was amplified with oligos containing BamHI and SalI restriction sites from the ATG by inverse PCR, followed by digestion with BamHI and SalI. On the other hand, pMK345 (Addgene Cat # 121179) and pMK348 (Addgene Cat # 121182) plasmids were digested with BamHI and SalI, and the fragment containing the antibiotic (Hygro and BSD respectively) was ligated to the HA containing plasmid. After confirmation of the sequence by sequencing, OsTIR1(F74G)-expressing HCT116 cells were transfected with both plasmids, followed by double antibiotic selection. Clones were then expanded and selected for PCR genotyping. Promising clones were checked microscopically (fluorescence), and were further subjected to FACS analysis and Western Blot.

## RNA interference
RNAi transfection was carried out using RNAiMAX following the manufacturer's instructions. U2OS cells were transfected using the following siRNAs for 48 or 72 h: si*LUC* (5′- CGU ACG CGG AAU ACU UCG ATT-3′), si*LMNA* (5′-CAG UCU GCU GAG AGG AAC ATT-3′), si*LAP2A* (5′-GAG AAU UGA UCA GUC UAA GTT-3), si*RECQ1* (5′-UUA CCA GUU ACC AGC AUU ATT-3′), si*PRIMPOL* (5′-GAG GAA ACC GUU GUC CUC AGU GUA U-3′) (all purchased from Microsynth), and si*KDM3* (ON-TARGETplus SMARTPool Cat # L-017301-00-0005).

## Biochemical fractionation, protein extraction and Western Blotting
Biochemical fractionation of cells was performed as previously described[68]. To determine the levels of depletion of the proteins of interest, protein extracts from all cell lines were prepared in Laemmli buffer (4% SDS, 20% glycerol, and 120 mM Tris- HCl, pH 6.8) and sonicated with a Bandelin Sonoplus Mini 20-System sonicator (3 pulses of 1.5 s, 70% amplitude). Equal amounts of protein (20 µg) were loaded onto 4%-15% gradient Mini-PROTEAN TGX Precast Protein Gels (BioRad). Proteins were separated by electrophoresis at 16 mA followed by transferring the proteins to Amersham nitrocellulose membranes (Merck) for 1-1.5 h at 350 mA at 4 °C in transfer buffer (25 mM Tris, 192 mM glycine) containing 20% methanol. Upon transferring, membranes were stained with Ponceau and imaged, followed by blocking in 5% milk in 0.1% TBST (TBS 1x supplemented with 0.1% Tween-20) for 1 h. Next, membranes were incubated in primary antibodies diluted in 5% milk/TBST overnight at 4 °C. eAf1521 in particular was diluted in 2% milk/TBST. Upon washing the membranes three times with 0.1% TBST, secondary antibodies were added for 45 min at room temperature. Membranes were then washed again three times with 0.1% TBST, followed by imaging in Fusion Solo (Vilber Lourmat) using ECL detection reagent or ECL Sirius. Proteins were quantified where necessary using ImageJ 64 software.

## Determination of PAR levels by CHAPS extracts and Western blot (for 96-10 and E6F6A antibodies specifically)

Cell pellets were lysed in 1 mL of lysis buffer (40 mM HEPES pH 7.5, 120 mM NaCl, 0.3% CHAPS) supplemented with cOmplete™ EDTA-free protease inhibitor cocktail (Sigma-Aldrich), 10 μM of PARP-1/2 inhibitor Talazoparib/BMN-673 (Selleckchem) and PARG inhibitor PDD00017272 (Tocris Bioscience) to block PAR turnover. Lysates were briefly sonicated for 30 sec on ice and mixed for 30 min on a rotating mixer in a cold room. Insoluble material and cellular debris were removed by centrifugation for 5 min at 1000 g. The supernatant was mixed with an equal volume of 4x Laemmli sample buffer (Bio-Rad) containing 5% β-mercaptoethanol. Lysates were resolved by 4–12% linear gradient SDS-PAGE (Bio-Rad) and transferred onto a 0.2 μm nitrocellulose membrane. PAR was revealed by Western blot using the anti-MAR/PAR antibody E6F6A or the anti-PAR antibody 96-10. The mouse monoclonal antibody clone C2-10 was used to detect PARP1. Lamin A was targeted with the anti-lamin A antibody L1213. Nonspecific antibody binding was blocked using 5% nonfat dried milk in PBS solution containing 0.1% Tween-20 (PBST). Primary antibodies were diluted in 5% milk in PBST and incubated overnight at room temperature on a rocking shaker. Prior to adding the secondary antibodies, the membranes were washed five times using PBST containing 5% milk. Horseradish peroxidase (HRP)-conjugated goat anti-rabbit secondary antibodies were allowed to bind at room temperature for 30 min. Membranes were washed in PBST and revealed using Western Lightning Plus-ECL enhanced chemiluminescence substrate (Revvity Health Sciences) according to the manufacturer's instructions and added to the membrane for 1 min. Excess substrate was removed prior to imaging on autoradiography films.

## Immunoprecipitation

For co-immunoprecipitation (IP), cells were scraped off the plate in IP buffer (20 mM Tris-HCl pH 7.5, 150 mM NaCl, 2 mM EGTA, 2 mM MgCl2, 0.5% NP-40, 1 mM DTT, 1x Complete EDTA-free Protease Inhibitor cocktail, Phosphatase inhibitor cocktail 2 and 3, PARGi 1 μM, and PARPi 10 μM) and incubated 10 min on ice. The samples were then sonicated with a Bandelin Sonoplus Mini 20-System sonicator (3 pulses of 1.5 s, 70% amplitude), followed by centrifugation for 10 min at 1300 g The soluble fractions were optionally treated with 50 U/μl of benzonase and used as input for the IP (1 mg/IP), followed by incubation with the Lamin A/C antibody (E1) (5 μg/IP) overnight at 4 °C, and blocking with BSA-blocked protein G dynabeads for 4 hr at 4 °C. Beads were washed three times in IP buffer without benzonase, followed by elution of complexes from the beads using SDS PAGE sample buffer. Samples were analyzed by SDS PAGE and immunoblotting.

## DNA fiber analysis

U2OS or HCT116 cells were cultivated asynchronously and subsequently labeled with two different thymidine analogs: 30 μM of chlorodeoxyuridine (CldU) for 25 min, followed by three washes with warm PBS 1x, and 250 μM of 5-iodo-2′-deoxyuridine (IdU) for 25 min alone or in combination with mild doses of genotoxic treatments (100 nM CPT or 20 nM ETP). To evaluate the impact of ADP-ribosylation on replication fork progression, U2OS cells were pre-incubated in DMEM containing 1 μM PARG inhibitors for two h before the initiation of the CldU labeling, which were retained during both CldU and IdU labeling. To evaluate the contribution of chromatin compaction on replication fork slowing, U2OS cells were pre-incubated in DMEM containing 1 μM G9a inhibitors which were maintained during both CldU and IdU labeling. In the HCT116 mAID2-LMNA cells, in order to induce degradation of Lamin A/C, 1 μM 5-Ph-IAA was added 24 h before the CldU/IdU pulse labeling and was maintained during the labeling. Upon IdU labeling, the cells were washed three times with cold PBS 1x, collected by standard trypsinization and resuspended in cold PBS 1x at $3 \times 10^5$ cells/mL. 3 μL of this cell suspension were then mixed with 7 μL of lysis buffer (200 mM Tris-HCl, pH 7.5, 50 mM EDTA, and 0.5% (w/v) SDS) on a glass slide placed horizontally. After an incubation of 6 min at RT, the slides were tilted at a 45° angle to stretch the DNA fibers onto the slide. The resulting DNA spreads were air-dried, fixed in ice-cold 3:1 methanol/acetic acid for 10 min, air-dried once more, and stored at 4 °C overnight. The following day, the DNA fibers were denatured by incubation in 2.5 M HCl for 1 h at RT, washed five times with PBS 1x and blocked with 2% BSA in PBST (PBS 1x supplemented with 0.05% Tween-20) for 40 min at RT. The newly replicated CldU and IdU tracks were then stained for 2.5 h at RT in a humidified chamber, using two different anti-BrdU antibodies recognizing CldU (1:300) and IdU (1:80), respectively. After washing five times with PBST, the slides were stained with anti-mouse Alexa-Fluor 488 (1:300) and anti-rat Cy3 (1:300) or anti-rat AlexaFluor 555 (1:300) secondary antibodies for 1 h at RT in the dark in a humidified chamber. After washing another five times with PBST, the slides were air-dried and then mounted in 13 μL Prolong Gold antifade reagent. Microscopy imaging was performed using a Leica DM6 B microscope (HCX PL APO 63x objective). To assess fork progression, the CldU and IdU track lengths of at least 100 fibers per sample were measured using the line tool in ImageJ software and analyzed into IdU/CldU ratio in Microsoft Excel. Graphical and statistical analysis was carried out using GraphPad Prism 10.

## Analysis of chromosome spreads

U2OS cells were transfected with siLUC, siLMNA or siLAP2A and treated with 100 nM CPT for 3 h. The genotoxic agent was removed by washing three times with PBS 1x and the cells were then released into fresh DMEM medium containing 200 ng/mL nocodazole for 16 h. Cells were collected at 48 h upon transfection, washed and resuspended in hypotonic solution (0.075 M KCl) for 20 min at 37 °C. Cells were then fixed with ice-cold fixation buffer (methanol:acetic acid, 3:1). The fixation step was repeated another two times. Cells were then dropped onto pre-hydrated glass slides and air-dried. The following day, slides were mounted with Vectashield medium containing DAPI. Microscopy imaging was performed using a Leica DM6 B microscope (HCX PL APO 63x objective) at 63x magnification equipped with a camera (model DFC360 FX; Leica) and visible chromosome abnormalities per metaphase spread were counted. Graphical and statistical analysis was carried out using GraphPad Prism 10.

## Proximity ligation assays (PLA)

**Lamin A/C:EdU PLA.** U2OS or HCT116 cells were asynchronously grown on sterile ibidi slides (ibidi, Cat # 80827). Cells were then treated with 100 nM CPT or 20 nM ETP for one h in total, followed by 25 μM EdU (10 min for the untreated cells, 12 min for the ETP-treated cells, 13 min for the CPT-treated cells) before the end of the one h. The cells were washed with PBS 1x and pre-extracted for 5 min using CSK-buffer (10 mM HEPES, 50 mM NaCl, 0.3 M Sucrose, 3 mM MgCl2, 1 mM EDTA or 1 mM EGTA, and 0.5% Triton X-100) on ice, followed by fixation in 4% formaldehyde at RT. Upon fixation, cells were washed three times with PBS 1x and permeabilized using 50 mM NH4Cl in 0.5% Triton X-100/PBS 1x for 3 min, followed by another 3 min in 0.5% Triton X-100/PBS 1x. Upon washing three times in PBS 1x, EdU detection was performed using a homemade Click reaction (0.1 M Tris pH 8.5, 0.1 M sodium ascorbate, 2 mM Cu2SO4 and 0.1 mM biotin-azide) for 1.5 h at 37 °C in a humidified chamber. After another three washes with PBS 1x, cells were incubated in blocking buffer at 37 °C for 1 h and incubated overnight at 4 °C with anti-Lamin A/C (Proteintech). After washing the primary antibody, cells were incubated with PLA probes for 1 h at 37 °C, ligation for 30 min 37 °C, and polymerase reaction for 100 min at 37 °C according to the manufacturer's instructions. After washing, cells were incubated at 37 °C for 30 min with secondary antibodies in blocking buffer containing DAPI (0.5 mg/mL). Following three washes in PBST and PBS 1x, the ibidi slides were kept in PBS 1x until being mounted with ibidi mounting medium only right before imaging acquisition.

Confocal imaging was performed using Leica SP8 inverse STED 3X and HC PL APO STED WHITE-motCORR 93x magnification. For confocal analysis, deconvolution was performed using Huygens Professional software. Image analysis and 3D-reconstruction was done using Imaris Software.

**PAR:EdU PLA.** U2OS or HCT116 cells were asynchronously grown on sterile 12-mm diameter glass coverslips coated with poly-L-lysine and were optionally pre-treated with 1 μM PARG inhibitors or 1 uM G9a inhibitors for two h. One h before fixation, cells were treated with 100 nM CPT (optionally in combination with PARGi or G9ai). The protocol was the same as the one for Lamin A/C: EdU with only the following modifications. The CSK buffer was supplemented with 10 μM PARPi and 1 μM PARGi and was used for 5 min on ice. The fixation was performed using 4% formaldehyde at RT, followed by MeOH for 5 mins in −20 °C. The permeabilization was performed for 6 min in 0.5% Triton X-100/PBS 1x. Upon secondary antibody and DAPI incubation, coverslips were washed twice in PBST and once in PBS 1x, and briefly immersed in distilled water, dried on 3 mm paper and mounted with Prolong Gold antifade reagent. Microscopy imaging was performed using a Leica DM6 B microscope (HCX PL APO 63x objective). PLA quantification was performed using an automated pipeline in Cell Profiler, whereas graphical and statistical analysis using GraphPad Prism 10.

**H3K9me3:EdU PLA.** HCT116-mAID2-mClover-LMNA cells were grown on sterile poly-lysine coated coverslips to be 60-70% confluent on the day of experiment. Cells were treated with 1 μM 5-Phenyl-1H-indole-3-acetic acid (5-Ph-IAA) for 24 h prior to the experiment to induce lamin A/C depletion. RPE-1 cells were grown on regular sterile glass coverslips. For Camptothecin (25 nM) and etoposide (20 nM) treated samples, cells were pulsed with EdU (10 μM) during the last 20 min of 1 h treatment. For Hydroxyurea, cells were labeled with EdU for 20 min before starting treatment. After treatment, cells were washed twice with cold 1x PBS, and were pre-extracted with 0.5% Triton in ice-cold cytoskeletal (CSK) buffer for 5 min at 4 °C and fixed with 4% Formaldehyde in PBS for 15 min at room temperature. After thorough washes with 1x PBS, cells were permeabilized with 0.1% Triton X-100 in PBS for 15 min room temperature. Samples were washed thoroughly with 1x PBS and then blocked with 5% BSA in PBS for 1 h at room temperature. EdU was conjugated with biotin azide (Jena Bioscience, CLK-1167-5) using copper-catalyzed Click chemistry for 1 h. Samples were then incubated with Rabbit Anti-H3K9me3 [EPR26601] (AB176916, Abcam) (1:1000 dilution in PBS,5% BSA) or Rabbit anti-G9a/EHMT2 antibody [C6H3] (3306, Cell signaling Technology) (1:50 dilution in PBS, 5% BSA) and Mouse Anti-biotin (AB_2339006, JacksonImmunoResearch) (1:1000 dilution in PBS,5% BSA) primary antibodies overnight at 4 °C. Proximity ligation was performed using Duolink PLA probes and Duolink Insitu Detection Reagent (Sigma) following manufacturer's protocol. EdU (biotin) was stained to detect S-phase cells using anti-Mouse af488 (Invitrogen, A-21206) (1:1000 dilution in PBS, 5% BSA). Nuclei were counter-stained with DAPI for 15 min at RT. Images were taken using Metafer 5 and PLA spot intensity (AU), the product of no. of spots and the mean intensity of spots per nucleus, was quantified using Metasystem.

**Electron microscopy**
U2OS cells were asynchronously grown and transfected with si*LUC*, si*LMNA* or si*LAP2A*. After 48 h of transfection and at 70–80% of confluency, cells were treated with 100 nM CPT for 1 h, followed by collection, resuspension in ice-cold PBS and crosslinking with 4,5′,8-trimethylpsoralen (10 μg/mL final concentration). Crosslinked cells were irradiated with pulses of UV 365 nm monochromatic light (UV Stratalinker 1800; Agilent Technologies). DNA was extracted according to Muzi-Falconi and Brown, 2018. Briefly, cells were lysed (1.28 M sucrose, 40 mM Tris-HCl [pH 7.5], 20 mM MgCl2, and 4% Triton X-100; Qiagen) and digested (800 mM guanidine-HCl, 30 mM Tris-HCl pH 8.0, 30 mM EDTA pH 8.0, 5% Tween-20, and 0.5% Triton X-100) at 50 °C for 2 h in presence of 1 mg/mL proteinase K. The DNA was purified using chloroform/isoamylalcohol (24:1) and precipitated in one volume of isopropanol. Finally, the DNA was washed with 70% EtOH and resuspended in 200 μL TE (Tris-EDTA) buffer. Restriction enzyme digestion followed (120 U of PvuII HF, New England Biolabs) in order to digest 6 μg of the purified genomic DNA for 5 h at 37 °C. RNase A (Sigma–Aldrich, R5503) to a final concentration of 250 ug/ml was added for the last 2 h of this incubation. The digested DNA was then purified using a Silica Bead Gel Extraction kit (Thermo Fisher Scientific) according to manufacturer's instructions. The Benzyl-dimethyl-alkyl-ammonium chloride (BAC) method was used to spread the DNA on carbon-coated 400-mesh nickel grids (G2400N, Plano Gmbh). Subsequently, DNA was coated with platinum using a High Vacuum Evaporator (EM BAF060, Leica) as described in Zellweger and Lopes (2018). The grids were imaged automatically at 28'000x using a Talos 120 transmission electron microscope (FEI; LaB6 filament, high tension ≤120 kV) with a bottom-mounted CMOS camera BM-Ceta (4096×4096 pixels) and the MAPS 3 software (Thermo Fisher Scientific). For the EM analysis, samples were annotated for replication intermediates using the MAPS offline viewer (V3.28, Thermo Fisher Scientific) and corresponding images were extracted. The replication intermediates were scored blind to the experimental condition using Fiji (Schindelin et al., 2012). For each experimental condition, at least 65 replication fork molecules were analyzed in two distinct biological replicates.

**Chromatin fiber analysis (ChromStretch)**
Chromatin fibers were prepared as described in ref. Gaggioli et al., NCB,2023 with minor changes. Following treatments, cells were harvested and washed with cold 1x PBS. Cells were lysed with 10 mM HEPES pH 7.9, 10 mM KCl, 1.5 mM MgCl2, 0.34 M sucrose, 10% glycerol, 1 mM DTT and protease inhibitor (cOmplete, mini, EDTA-free Protease Inhibitor Cocktail, Roche) for 5 min on ice. Samples were centrifuged (1,500 g for 5 min) at 4 °C to collect the released nuclei. Nuclei were resuspended in hypotonic buffer (3 mM EDTA, 0.2 mM egtazic acid, 1 mM DTT and protease inhibitor), spotted on Superfrost microscope slides and incubated in a humid chamber. Excess buffer was removed and the slides were allowed to dry for a maximum of 5 min. It was then transferred to a chamber containing lysis buffer pH 7 and incubated for 10–20 min. Chromatin fibers were isolated by letting the lysis buffer flow out of the chamber at a constant flow rate to facilitate stretching with the help of an equipment developed in the lab. Stretched chromatin fibers were fixed with 2% Formaldehyde in PBS for 15 min at room temperature. Slides were washed thoroughly in PBS and EdU was fluorescently labeled with Alexa Fluor 594 azide using Click chemistry. Slides were washed with PBS, blocked in 5% BSA in PBS for 1 h and then incubated with Rabbit anti-H3K9me3 [EPR26601] (ab176916, Abcam, 1:1000 in 5% BSA PBS) or Mouse anti-H3K9me3 [EPR26601] (ab317790, Abcam, 1:1000 in 5% BSA) and Rabbit anti-H3 (ab1791, 1:1,000 in 5% BSA PBS for 1 h) or Mouse anti-H3 (Cell signaling technology, 14269S) overnight at 4 °C. The primary antibodies were labeled with an anti-Rabbit secondary antibody conjugated to Alexa Fluor 488 (1:1000 in 5% BSA PBS) or Alexa Fluor 405 (1:1000 in 5% BSA PBS) for 1 h at room temperature and anti-Rabbit or anti-Mouse antibody conjugated to Alexa Fluor647 (1:1000 in 5% BSA PBS) for 1 h at room temperature. Chromatin fibers were imaged using a Leica ST5 confocal microscope equipped with an oil immersion 63× (HC PL APO CS2, NA 1.4) objective. H3K9me3 and H3 signal at EdU spots were quantified using ImageJ.

**Normalization of H3 signal in chromatin fibers.** As the role of the nuclear lamina/Lamin A in regulating replication-coupled chromatin assembly is well established (PMID: 34788845; 31883795), Lamin

depletion leads to global changes in histone levels around replication forks. To accurately assess histone density at replication forks under replication stress and/or Lamin A depletion, we normalized H3 levels using a spatial internal control. Specifically, we selected a fixed-size area immediately flanking the EdU-labeled replication bubble, on both the right and left sides, and calculated the average H3 signal in those regions. The normalized H3 level at the EdU-labeled bubble was then determined using the following ratio:

Normalized H3 = H3 signal at EdU / (Average H3 signal at Right + Left flanks).

### Flow cytometric analysis (EdU/ γH2AX/DAPI)
U2OS or HCT116 WT and degron cell lines were labeled with 10 µM EdU for 30 min, harvested by standard trypsinization and subsequently fixed for 10 min in 4% formaldehyde/PBS 1x. Cells were then washed twice and blocked over night at 4 °C with 1% BSA/PBS 1x, pH 7.4. Next, they were permeabilized with 0.5% saponin/1% BSA/PBS 1x, and stained with primary mouse anti-γH2AX antibody diluted at 1:1000 in 0.5% saponin/1% BSA/PBS 1x for 2 hr. This was followed by incubation with a Goat anti-mouse Alexa 647 antibody diluted at 1:125 in 0.5% saponin/1% BSA/PBS 1x for 30 min. The incorporated EdU was labeled according to the manufacturer's instructions. Total DNA was stained with 1 µg/mL DAPI dissolved in 1% BSA/PBS 1x. Samples were measured on an Attune NxT Flow Cytometer (Thermo Fisher) and analyzed using FlowJo software V.10.0.8 (FlowJo, LLC).

### Reporting summary
Further information on research design is available in the Nature Portfolio Reporting Summary linked to this article.

## Data availability
Raw data used to generate all graphs and derived statistics are provided in the Source data table. Original, uncropped blots can be found in the Source data_blots file. All original microscopy images are far too numerous and large to be stably uploaded on a public repository, but will be made available upon request. Source data are provided with this paper.

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

## Acknowledgements

We are grateful to Jana Döhner, Nicolas Schilling, Flurin Sturzenegger and the whole Center for Microscopy of the University of Zurich for technical assistance with microscopy and image analysis. We also thank Nana Naetar, Lukas Muskalla and David Kubon for technical support. We are particularly grateful to Roland Foisner and Michael Hottiger for important reagents, technical and conceptual contributions, and critical reading of the manuscript. We are grateful to all members of the Lopes lab for useful discussions and suggestions on the manuscript. Work in the Lopes lab was supported by the SNF Project grants 310030_189206 and 310030_219393. V.C. received a State Scholarships Foundation "N.D. Chrisovergis" Bequest for Graduate studies abroad. Work in the Taneja lab was supported by the Oncode Institute, which is partly financed by the Dutch Cancer Society and Vidi funding (project no. 114122) and ERC funding (grant no. ChOReS, 101078750/ #114168). We thank the Josephine Nefkens Cancer Program for infrastructure support to the Taneja lab.

## Author contributions

V.C. obtained new cell lines, performed most DNA fiber, IF/PLA, WB, FACS and EM experiments, and all IPs and metaphase spreads. In the lab of N.T., J.P. performed PLA and ChromStretch experiments on H3K9me3, while C.B. performed PLA experiments on G9a. D.G.-A.

designed and assisted with cell line generation, and a subset of DNA fiber analysis. P.U.-C. contributed with a subset of DNA fiber and FACS, biochemical fractionations, WBs and EM experiments. M.R. performed DNA fiber and WB experiments upon *KDM3A* downregulation. M.A. assisted in sample processing and image analysis for EM experiments. S.A. contributed with DNA fiber and WB experiments upon *LAP2A* downregulation together with V.C. and P. U.-C. J.-P. G. performed a subset of WBs to detect global PAR levels, in G. G. P.'s lab. M. L. designed and supervised the project, with crucial contributions of V.C. and N.T.M.L., and V.C. wrote the manuscript, which was finalized with assistance of D.G.-A. and contributions of all coauthors.

## Competing interests
N.T. holds an international patent for ChromStretch technology filed under PCT/NL2023/050120. No other authors have competing interests.
