## [Transparent Peer Review file · Nature Communications]

Nucleoplasmic Lamin A/C controls replication fork restart upon stress by modulating local H3K9me3 and ADP-ribosylation levels

Corresponding Author: Professor Massimo Lopes

Version 0:

Reviewer comments:

Reviewer #1

(Remarks to the Author)

The manuscript explores the role of Lamin A/C in modulating mild replication stress (RS) during cancer chemotherapy. Mild RS, a common strategy in chemotherapy, depends on mechanisms like fork slowing, reversal, and restart to maintain genomic stability. While the specific roles of various proteins and structures in this process are not fully understood, this study investigates Lamin A/C's involvement in RS response. The authors demonstrate that Lamin A/C, in conjunction with its nucleoplasmic partner LAP2a, is essential for inducing active fork slowing and preserving chromosome stability under mild genotoxic stress. Inactivating Lamin A/C disrupts poly-ADP ribosylation (PAR) at replication sites, which in turn leads to deregulated fork restart, driven by the helicase RECQ1. Furthermore, the study uncovers that the heterochromatin mark H3K9me3 accumulates in response to mild RS and that Lamin A/C is required for preventing its premature removal by the demethylase JMJD1A/KDM3A. Loss of H3K9me3 phenocopies Lamin A/C inactivation, suggesting that these three factors—Lamin A/C, H3K9me3, and PARylation—are key modulators of replication fork dynamics and stability. The results have significant implications for understanding chemotherapy responses and the role of Lamin A/C deregulation in disease.

This manuscript presents solid and comprehensive work that advances our understanding of the molecular mechanisms governing replication stress tolerance and stability. The identification of Lamin A/C as a central player in regulating replication fork dynamics, through its interaction with H3K9me3 and PARylation, is a novel contribution that ties together various aspects of the replication stress response. The manuscript is well-written, with clear experimental details and insightful conclusions.

Some discussion points or clarifications which might strengthen the MS:

-The manuscript extensively discusses replication stress and fork dynamics, largely relying on proximity ligation assays (PLA), which are sensitive to EdU incorporation levels. However, it would be helpful to provide more detailed information regarding EdU incorporation levels in the presence of CPT and Etp. How do the authors interpret changes in EdU signal intensity or foci number in response to these genotoxic agents, especially in relation to the observed alterations in PARylation and RECQ1 activity? Clarifying or expanding on this point would offer a deeper understanding of how EdU incorporation correlates with the observed molecular events and improve the interpretation of the PLA data.

-There is an apparent discrepancy between the data presented in Extended Figure 4b and 4c. In Figure 4b, there is an increase in poly-ADP ribosylation (PAR) levels, but the authors do not mention this observation in the main text or figure legend. It would be helpful to provide a discussion of this increase in PAR and its potential significance in the context of Lamin A/C inactivation. Could the authors clarify this point and discuss the implications of increased PAR in relation to their findings?

-In the proximity ligation assay (PLA) between , the authors switch from reporting foci number to foci intensity (H3K9me3 and EdU). This change is somewhat unexpected, and the rationale behind this shift should be explained. Is there a specific reason for using foci number rather than intensity in these analyses, or would it be possible to maintain consistency in reporting both foci intensity and number? A brief discussion on the criteria for this choice would enhance clarity.

-In several figures, the authors interchange the use of camptothecin (CPT) and etoposide (Etp) without clear explanation. Could the authors provide a rationale for this choice, or would it be preferable to focus on one agent to avoid potential confusion?

-The data shown in Extended Figure 4b could be more informative if the authors provide the merge of the images to view the extent of colocalization between the two signals.

Reviewer #2

(Remarks to the Author)

Comments:

The manuscript Veronica Cherdyntseva et al., addresses an important biological question, the function of Lamin A/C upon mild replication stress. To induce mild replication stress, throughout the study, the authors describe low dose of camptothecin and etoposide (Topoisomerase I and II inhibitors). The authors use HCT116 and U2OS cell lines for their entire study.

Using EdU:Lamin A/C proximity ligation assay (PLA), first, the authors convincingly show that LaminA/C dynamically interacts with the replication factories and this interaction is reduced upon mild replication stress induction. They also show that LaminA/C's nucleoplasmic interactor LAP2alpha downregulation reduced LaminA/C:EdU PLA foci.

Second, using siRNA and auxin-inducible degradation approach, the authors found that the inactivation of LaminA & LAP2alpha led to unrestrained fork progression in the presence of replication stress. This unrestrained fork progression was associated with increased chromosome breaks (Lamin A downregulation + Mild replication stress background).

Third, the authors explored the reason for the unrestrained fork progression in LaminA/C-Lap2alpha defective cells in the presence of mild replication stress. They initially ruled out the involvement of PRIMPOL. Later using EM visualisation and DNA fibre assay they conclude that the deregulation in RECQ1 helicase activity is responsible for fork progression (Lamin A downregulation + Mild replication stress background). In other words, LaminA/C-Lap2alpha functions by slowing down replication forks by keeping reversed forks levels relatively high by negatively regulating RECQ1's fork restart activity (In replication stress background).

Fourth, since it is known previously that auto-PARylated PARP1 interact with RECQ1 (Matteo Berti et al., 2013) and prevents the restart of reversed forks, the authors speculated that LaminA/C might promote PARylation globally to control RECQ1 activity. By stabilising global PAR levels by the addition of PARG inhibitors, the authors demonstrate the suppression of unrestrained fork progression in LaminA and LAP2alpha deficient cells.

Fifth, next the authors move onto the chromatin context questioning whether the heterochromatin marks/chromatin modifications might have a role on LaminA/C regulated fork restart. They initially show that the lysine methyl transferase G9a associate with the nascent DNA in the presence of replication stress and stalling with increased heterochromatic signature. Using G9a specific inhibitor, they demonstrate an unrestrained fork progression upon replication stress which is dependent on RECQ1 but not Primpol.

In line with the chromatin context, finally, using Chromstreich approach (Vincent Gaggioli et al., 2023) the authors demonstrate reduced H3K9me3 marks in LaminA/C deficient cells, suggesting a link between LaminA/C – Heterochromatin Marks and Replication fork restart.

Overall, the findings provide an important contribution in the knowledge of LaminA/C and replication stress. Though the data support the conclusion and the final model, I have several concerns in the organisation of the manuscript, concentrations of drugs used, controls used and in terms of relating to the previously published evidence on fork reversal and restart. Please find below my specific concerns and suggestions that need to be addressed.

Major Concerns:

1) In this manuscript, the authors use 100 nM CPT and 20 nM ETP for describing mild replication stress. In Figure 1, authors refer their previous work Ralph Zellweger et al., 2015 JCB where 25 nM CPT is used. It is important that the authors show pRPA, pChk1 and Gamma H2AX with gradient concentration of CPT until 100 nM or beyond to rule out any detectable double strand breaks (DSB's). I agree for ETP concentration used in this manuscript, it is clear from Ralph Zellweger et al., 2015 JCB that there is no DSB's and it reflects mild replication stress.

2) My concern with 100 nM CPT is, i) in previous observations from Matteo Berti et al., 2014 NSMB, in Figure 5a showed DSB's induction using PFGE with 100 nM of CPT. ii) Other observation from Arnab Ray Chaudhuri et al 2012 NSMB, in Figure 5a/b showed a mild increase in DSB's in 100 nM and a significant increase in Gamma-H2AX foci. Here, the authors, in the context of CPT might be looking at DSB's as well and not mild replication stress alone.

3) In the introduction, the authors refer Mayank Singh et al., 2013 Mol Cell Biol and Simona Graziano et al., 2021 J Biol Chem that LaminA/C role in recruiting RPA and Rad51 to stalled replication forks. Given that si Lamin A reduced reversed forks in Figure 3A, and the authors lab have shown previously Rad51's role in reversed fork formation (Ralph Zellweger et al., 2015 JCB), it is important to monitor the level of Rad51 in the presence of mild replication stress induced by CPT and

ETP.

4) In Figure 3, in addition to the fork restart efficiency and reversed fork levels, scoring ssDNA at gap junction (similar to Simona Graziano et al., 2021 J Biol Chem) might help to understand the correlation between ssDNA gaps, fork reversal and fork restart upon mild replication stress.

5). For Figure 3 c & f and 3 i & j, it is important to include RECQ1 (K119R), ATPase deficient mutant as control both for fibre restart and EM. This will help in this part to conclude role of ATPase activity of RECQ1 in fork restart (similar to Matteo Berti et al., 2013) in LaminA and LAP2alpha background.

6) I acknowledge the tedious work flow of EM but it will be useful if EM reversed forks could be scored for Figure 4i and 5i.

Reviewer #3

(Remarks to the Author)

The manuscript examines the activities of Lamin A/C, histone methylation, and ADP-ribosylation in controlling replication fork dynamics in response to DNA damage. A previous publication found that LaminA/C is important for replication fork stability and restart. That work did not find any defect in fork reversal. The authors explain that limitations in other studies was the use of genetic systems with prolonged inactivation of LaminA/C which could cause cells to adapt or otherwise mask a phenotype. Therefore, they want to examine the consequences of “acute” inactivation. They use both siRNAs and a degron system to inactivate Lamin A/C and use DNA fiber analysis of replication kinetics and EM analysis of replication intermediates to examine replication. They find that inactivating these proteins causes unrestrained fork progression and increased chromosome abnormalities in the presence of DNA damage. Based on localization studies with proximity ligation assays and the similar phenotype of LAP2alpha inactivation, they conclude that nucleoplasmic Lamin A/C is associated with sites of replication and mediates fork slowing and fork stabilization. Furthermore, they find that inactivating Lamin A/C reduces poly-ADP-ribosylation (PAR) near newly synthesized (nascent) DNA but does not alter global cellular levels of PAR. This is important since PAR was previously shown to inhibit the fork restart helicase RECQ1. Thus, they suggest that RECQ1 is not inhibited properly in the absence of Lamin A/C. This causes inappropriate fork restart, a reduction in fork reversal, and the faster fork speeds. Furthermore, they link these phenotypes to a lack of proper H3K9me3 accumulation near the stressed replication forks which prevents proper local chromatin compaction. They conclude that Lamin A modulates local chromatin compaction which then increases PARylation which inhibits RECQ1. These results contrast with the previous studies that reported that Lamin A was required for fork restart and fork protection via regulation of RPA and RAD51. Furthermore, the previous study indicated there was no fork reversal defect in Lamin A-deficient cells (Graziano et al., JBC 2021).

Overall, I found the data interesting but several of the conclusions are not sufficiently justified. I outline some specific concerns below.

1) The localization data is based on a proximity ligation assay between Lamin A/C and EdU. This assay could return a positive result for any abundant nucleoplasmic protein. To ensure there is actual enrichment of Lamin A/C at replication factories, the authors should do an experiment in which EdU is added and then removed for some time to label DNA that is not at replication factories. If there is selective localization of Lamin A/C at replication factories then the PLA signal should no longer be visible if the EdU is no longer near the replication factory.

2) It is not surprising that the amount of PLA between Lamin A/C and EdU would be decreased when cells are first treated with DNA damaging agents that slow replication since the amount of EdU incorporated would decrease due to slower elongation and decreased origin firing. Therefore, the conclusion that localization decreases in response to replication stress is not justified by the experiment in Figure 1e.

3) A key aspect of this report compared to prior studies is the use of “acute” depletion of Lamin A. This is used as one rationale to explain differences in results. Further studies on this point are needed.

a. Does chronic inactivation of Lamin A yield different results in these author's hands? This could be done by depleting Lamin A with the degron system for multiple days or even a couple of weeks.

b. The AID2 degron system often can yield complete protein degradation in 1-2 hours. The investigators only show how much is depleted at a 24h time point and their experiments were done at that time. Many secondary effects could have happened in 24 hours. Why not do even shorter time points like 2 hours? Furthermore, the authors extensively make use of siRNA treatment for 48 hours which they also consider “acute” depletion. It is unclear why some experiments are only performed with the siRNA and some with the degron system. For example, the chromosomal abnormalities in Figure 2 and the unrestrained fork progression analysis and fork reversal analysis in figure 3 are done with siRNA. Are the same results obtained with a short degradation of LMNA protein? The important analysis of ADPr levels is also only done with the siRNA. Why switch between systems? I suggest the authors complete key experiments using both the degron and siRNA. Otherwise the reader is left wondering if the other system yielded different results.

4) Extended data 2c is used to show that there is no “marked alteration” of cell cycle progression after LMNA degradation. However, 5h after addition of auxin to clone 13 there is over a 10% increase in EdU positive cells and 24h there is a significant number of cells with between 2n and 4n DNA content which are EdU negative.

5) I don't understand how G9a inhibition (and subsequent changes in local chromatin) would affect the level of PAR at replication forks. Does it affect PARP recruitment to forks? Does it affect the amount of CPT-induced DNA damage at forks? Does it alter the activity of PARP1/2? Does it alter the activity or levels of PARG? The previous report by one of these

authors found an increase in PARG at forks in response to G9a inhibition and reduction in RECQ1 (Gaggioli et al., NCB 2023). Are those differences underlying these effects?

6) The previous paper also showed that G9a inhibition caused fork degradation implying that it did not prevent fork reversal (Gaggioli et al., NCB 2023). How does that fit with the model that G9a promotes reversal and fork slowing?

7) The authors make conclusions about local chromatin "compaction". But they do not measure compaction in any experiment. Measurements of a single methylation mark seems insufficient to make conclusions about the chromatin state.

8) How do the authors reconcile their conclusion in this manuscript that PARG inhibition does not affect fork progression with their previous paper that reported that PARG inactivation causes replication fork slowing (Chaudhuri et al., MCB 2015)?

Version 1:

Reviewer comments:

Reviewer #1

(Remarks to the Author)

The authors took care of the reviewers's comments and gave comprehensive answers. I therefore recommend publication.

Reviewer #2

(Remarks to the Author)

The authors have satisfactorily addressed the comments raised in my initial review. I find that all major concerns have been appropriately resolved, and the manuscript has improved as a result. I have no further suggestions for revision at this stage and recommend that the manuscript be accepted for publication.

Reviewer #3

(Remarks to the Author)

I have no further comments for the authors.

Reviewer #1 (Remarks to the Author):

The manuscript explores the role of Lamin A/C in modulating mild replication stress (RS) during cancer chemotherapy. Mild RS, a common strategy in chemotherapy, depends on mechanisms like fork slowing, reversal, and restart to maintain genomic stability. While the specific roles of various proteins and structures in this process are not fully understood, this study investigates Lamin A/C's involvement in RS response. The authors demonstrate that Lamin A/C, in conjunction with its nucleoplasmic partner LAP2a, is essential for inducing active fork slowing and preserving chromosome stability under mild genotoxic stress. Inactivating Lamin A/C disrupts poly-ADP ribosylation (PAR) at replication sites, which in turn leads to deregulated fork restart, driven by the helicase RECQ1. Furthermore, the study uncovers that the heterochromatin mark H3K9me3 accumulates in response to mild RS and that Lamin A/C is required for preventing its premature removal by the demethylase JMJD1A/KDM3A. Loss of H3K9me3 phenocopies Lamin A/C inactivation, suggesting that these three factors—Lamin A/C, H3K9me3, and PARylation—are key modulators of replication fork dynamics and stability. The results have significant implications for understanding chemotherapy responses and the role of Lamin A/C deregulation in disease.

This manuscript presents solid and comprehensive work that advances our understanding of the molecular mechanisms governing replication stress tolerance and stability. The identification of Lamin A/C as a central player in regulating replication fork dynamics, through its interaction with H3K9me3 and PARylation, is a novel contribution that ties together various aspects of the replication stress response. The manuscript is well-written, with clear experimental details and insightful conclusions.

We are grateful to this reviewer for sharing our views on the relevance and novelty of the reported findings. In the attempt to further improve the manuscript, we have addressed specific criticisms and suggestions here below.

Some discussion points or clarifications which might strengthen the MS:

1-The manuscript extensively discusses replication stress and fork dynamics, largely relying on proximity ligation assays (PLA), which are sensitive to EdU incorporation levels. However, it would be helpful to provide more detailed information regarding EdU incorporation levels in the presence of CPT and Etp. How do the authors interpret changes in EdU signal intensity or foci number in response to these genotoxic agents, especially in relation to the observed alterations in PARylation and RECQ1 activity? Clarifying or expanding on this point would offer a deeper understanding of how EdU incorporation correlates with the observed molecular events and improve the interpretation of the PLA data.

We thank this reviewer for raising this important point (shared with reviewer 3, point 2) and apologize for not sufficiently stress in the submitted manuscript how we addressed this potential issue. Indeed, PLA signals for proximity to nascent DNA are sensitive to EdU incorporation levels, which are expected to change when cells are treated with genotoxic agents (albeit in very mild doses, as those used in this work). To avoid this issue, we have performed pilot experiments and adapted EdU incorporation times upon treatment, in order to obtain comparable EdU incorporation signals in all conditions and allow reliable PLA measurements. In the revised manuscript, we show these comparable EdU incorporation levels – which were observed with minimal variations throughout our experiments – by displaying a representative PLA experiment in both Ext Data Figs. 1(c) and 4(e), while presenting and discussing in the text how Lamin- or PAR-levels at forks are affected upon specific treatments and/or specific genetic perturbations (Fig. 1e and Fig. 4c, respectively).

2-There is an apparent discrepancy between the data presented in Extended Figure 4b and 4c. In Figure 4b, there is an increase in poly-ADP ribosylation (PAR) levels, but the authors do not mention this observation in the main text or figure legend. It would be helpful to provide a discussion of this increase in PAR and its potential significance in the context of Lamin A/C inactivation. Could the authors clarify this point and discuss the implications of increased PAR in relation to their findings?

We have also noticed the slight increase in total PAR levels upon Lamin inactivation, which is visible in Ext Data Fig. 4b. However, this mild effect was not reproducible in all biological replicates of the same experiment and/or when PAR levels were detected by a different protocol and different detection reagents (i.e. Ext Data Fig. 4c). This most likely reflects the detection of different types/subpopulations of PAR chains by the different reagents, or important differences in the stability of specific PAR chains upon different protein extraction procedures. Exactly for this reason, we concluded that the striking replication defects associated with Lamin A/C inactivation are unlikely to reflect general effects on global PAR levels in the cells, and focused follow-up experiments on the specific PAR-events occurring at replication forks (PAR-EdU PLA). In light of the marked correlation we observed throughout the manuscript between PAR-EdU signals and replication fork defects, we decided to only include these (overall negative) results on total PAR levels in the Extended Data, and would find unnecessary and potentially confusing to include further discussion of these data in the main text.

3-In the proximity ligation assay (PLA) between , the authors switch from reporting foci number to foci intensity (H3K9me3 and EdU). This change is somewhat unexpected, and the rationale behind this shift should be explained. Is there a specific reason for using foci number rather than intensity in these analyses, or would it be

possible to maintain consistency in reporting both foci intensity and number? A brief discussion on the criteria for this choice would enhance clarity.

We thank the reviewer for this important remark. As it is also clear from the materials and methods, PLA experiments were performed in two different labs (Lopes/Taneja), where specific protocols and analysis pipelines have been previously optimised. Lamin-EdU and PAR-EdU PLAs assays were optimized in the Lopes lab and yielded the most reliable results when scored based on PLA foci numbers after short EdU labelling pulses.

However, assessing the levels of epigenetic writers (G9a) or histone marks (H3K9me3) at replication factories typically requires extended EdU labelling to capture sufficient DNA (~40 kb with a 20-minute pulse), reflecting chromatin modifications over approximately 200–300 nucleosomes, depending on the degree of chromatin compaction behind the fork. Under these conditions, quantification based on total PLA signal intensity is more appropriate than foci number. Since heterochromatin-associated foci tend to cluster, relying on PLA foci counts may underestimate the actual signal. Therefore, consistent with the approach described in Gaggioli et al. NCB 2023, we used total spot intensity as a more robust and quantitative readout, which also correlated well with our single-molecule chromatin fiber data that represents quantification from individual replication bubbles.

Nevertheless, to address the reviewer's concern, we have now included in the Extended Data representative PLA experiments from Fig. 1e and Fig 4c, re-analyzed using total signal intensity (see Ext Data Figs. 1d and 4f). These analyses confirm the trends observed using foci counts in the main figures. The manuscript text has been updated accordingly.

4-In several figures, the authors interchange the use of camptothecin (CPT) and etoposide (Etp) without clear explanation. Could the authors provide a rationale for this choice, or would it be preferable to focus on one agent to avoid potential confusion?

We have now discussed in the opening section of the Results why treatment with CPT (mild fork slowing, minimal levels of DSBs) and ETP (mild fork slowing, no detectable DSBs) have been used in parallel for a few key experiments. These include the control of active fork slowing by DNA fibers by Lamin and G9a (Figs. 2h and 5f) and G9a/H3K9me3-EdU PLA assays (Fig. 5a-d). Once we established that the two treatments induce comparable effects for these key readouts, we extended our analysis to additional genetic and chemical perturbations, as well as to more sensitive or time/cost-demanding approaches (EM, ChromStretch, other PLAs) by selecting the one genotoxic agent that had been mainly used in previous experiments, within this project and other projects in our labs (i.e. CPT for EM analysis and ChromStretch, ETP for additional fibers assays on the role of RECQ1, Primpol, PARG).

5-The data shown in Extended Figure 4b could be more informative if the authors provide the merge of the images to view the extend of colocalization between the two signals.

The reviewer most likely refers to Fig. 4b (there is no IF in Ext Data Fig. 4b). If this is the case, we note that the PLA signal is already *per se* the most reliable readout of the "colocalization" that we aimed to measure in this experiment, as it indirectly measures the close proximity of PAR chains to a subset of replication factories. In that respect, we are not sure that showing a merge of the signal would be a significant addition to the figure.

Reviewer #2 (Remarks to the Author):

Comments:

The manuscript Veronica Cherdyntseva et al., addresses an important biological question, the function of Lamin A/C upon mild replication stress. To induce mild replication stress, throughout the study, the authors describe low dose of camptothecin and etoposide (Topoisomerase I and II inhibitors). The authors use HCT116 and U2OS cell lines for their entire study.

Using EdU:Lamin A/C proximity ligation assay (PLA), first, the authors convincingly show that LaminA/C dynamically interacts with the replication factories and this interaction is reduced upon mild replication stress induction. They also show that LaminA/C's nucleoplasmic interactor LAP2alpha downregulation reduced LaminA/C:EdU PLA foci.

Second, using siRNA and auxin-inducible degradation approach, the authors found that the inactivation of LaminA & LAP2alpha led to unrestrained fork progression in the presence of replication stress. This unrestrained fork progression was associated with increased chromosome breaks (Lamin A downregulation + Mild replication stress background).

Third, the authors explored the reason for the unrestrained fork progression in LaminA/C-Lap2alpha defective cells in the presence of mild replication stress. They initially ruled out the involvement of PRIMPOL. Later using EM visualisation and DNA fibre assay they conclude that the deregulation in RECQ1 helicase activity is responsible for fork progression (Lamin A downregulation + Mild replication stress background). In other words, LaminA/C-Lap2alpha functions by slowing down replication forks by keeping reversed forks levels relatively high by negatively regulating RECQ1's fork restart activity (In replication stress background).

Fourth, since it is known previously that auto-PARylated PARP1 interact with RECQ1 (Matteo Berti et al., 2013) and prevents the restart of reversed forks, the authors speculated that LaminA/C might promote PARylation globally to control RECQ1 activity. By stabilising global PAR levels by the addition of PARG inhibitors, the authors demonstrate the suppression of unrestrained fork progression in LaminA and LAP2alpha deficient cells.

Fifth, next the authors move onto the chromatin context questioning whether the heterochromatin marks/chromatin modifications might have a role on LaminA/C regulated fork restart. They initially show that the lysine methyl transferase G9a associate with the nascent DNA in the presence of replication stress and stalling with increased heterochromatic signature. Using G9a specific inhibitor, they demonstrate an unrestrained fork progression upon replication stress which is dependent on RECQ1 but not Primpol.

In line with the chromatin context, finally, using Chromstreich approach (Vincent Gaggioli et al., 2023) the authors demonstrate reduced H3K9me3 marks in LaminA/C deficient cells, suggesting a link between LaminA/C – Heterochromatin Marks and Replication fork restart.

Overall, the findings provide an important contribution in the knowledge of LaminA/C and replication stress. Though the data support the conclusion and the final model, I have several concerns in the organisation of the manuscript, concentrations of drugs used, controls used and in terms of relating to the previously published evidence on fork reversal and restart. Please find below my specific concerns and suggestions that need to be addressed.

We thank this reviewer for the overall positive feedback on our work. We have tried to address all specific concerns here below.

Major Concerns:

1) In this manuscript, the authors use 100 nM CPT and 20 nM ETP for describing mild replication stress. In Figure 1, authors refer their previous work Ralph Zellweger et al., 2015 JCB where 25 nM CPT is used. It is important that the authors show pRPA, pChk1 and Gamma H2AX with gradient concentration of CPT until 100 nM or beyond to rule out any detectable double strand breaks (DSB's). I agree for ETP concentration used in this manuscript, it is clear from Ralph Zellweger et al., 2015 JCB that there is no DSB's and it reflects mild replication stress.

This point and the next one are addressed jointly just below.

2) My concern with 100 nM CPT is, i) in previous observations from Matteo Berti et al., 2014 NSMB, in Figure 5a showed DSB's induction using PFGE with 100 nM of CPT. ii) Other observation from Arnab Ray Chaudhuri et al 2012 NSMB, in Figure 5a/b showed a mild increase in DSB's in 100 nM and a significant increase in Gamma-H2AX foci. Here, the authors, in the context of CPT might be looking at DSB's as well and not mild replication stress alone.

We agree with the reviewer's concerns regarding the presence of minor levels of DSBs upon treatment with CPT 100nM and apologize if the submitted manuscript lacked clarity on this important point. In fact, this was the main reason for us to perform key experiments in this work with both treatments, which induce comparable effects in terms of global/active fork slowing, combined (CPT) or not (ETP) with low levels of DSBs. All our data strongly suggest that the entire pathway we describe (Lamin/LAP2 α – H3K9me3 – ADPr – RECQ1) is required to modulate fork progression, remodelling and restart even in conditions that do not induce any detectable DNA breakage and do not reportedly activate any detectable DNA damage response. We have now clarified this important point in the opening section of the Results (see also our response to Reviewer 1, point 4). We are confident that the revised manuscript clarifies this point and conveys the messages more clearly.

3) In the introduction, the authors refer Mayank Singh et al., 2013 Mol Cell Biol and Simona Graziano et al., 2021 J Biol Chem that LaminA/C role in recruiting RPA and Rad51 to stalled replication forks. Given that si Lamin A reduced reversed forks in Figure 3A, and the authors lab have shown previously Rad51's role in reversed fork formation (Ralph Zellweger et al., 2015 JCB), it is important to monitor the level of Rad51 in the presence of mild replication stress induced by CPT and ETP.

This point and the next one are addressed jointly just below.

4) In Figure 3, in addition to the fork restart efficiency and reversed fork levels, scoring ssDNA at gap junction (similar to Simona Graziano et al., 2021 J Biol Chem) might help to understand the correlation between ssDNA gaps, fork reversal and fork restart upon mild replication stress.

We thank the reviewer for these important remarks. We are aware of the mentioned reports, which are indeed cited in our manuscript. We note however that the observations in those reports were collected upon prolonged fork stalling by HU-induced nucleotide-depletion. All evidence collected in our work suggests that, upon the mild genotoxic treatments we used, compatible with residual fork progression, the role of Lamin A/C in modulating fork progression reflects a specific mechanism of regulation of RECQ1-mediated restart, rather than a role in promoting the formation of reversed replication forks. To consolidate this point, we did assess a) the size of ssDNA regions detectable by EM at replication forks (a key intermediate previously implicated in fork reversal), b) the chromatin loading of the ssDNA binding proteins RPA and RAD51 (as RAD51 loading was shown to drive reversed fork formation). In our experimental conditions (very mild CPT or ETP treatments), ssDNA accumulation at forks is not affected by Lamin A/C or LAP2 α inactivation. Moreover, RAD51 chromatin loading is practically undetectable and anyway unchanged upon Lamin A/C depletion, consolidating our key conclusion that the defects in fork progression and remodelling reflect more downstream events in fork remodelling and restart. These new important (albeit negative) observations have been included and briefly discussed in the revised manuscript (Extended Data Fig. 3h-i).

5). For Figure 3 c & f and 3 i & j, it is important to include RECQ1 (K119R), ATPase deficient mutant as control both for fibre restart and EM. This will help in this part to conclude role of ATPase activity of RECQ1 in fork restart (similar to Matteo Berti et al., 2013) in LaminA and LAP2alpha background.

We note that the functional role of RECQ1 and its ATPase activity in the restart of reversed replication forks has been extensively characterized in previous studies. Key observations in our manuscript already establish the relevance of RECQ1-mediated fork restart for how Lamin A/C and LAP2a control fork progression and remodelling upon mild genotoxic stress (Fig. 3a-j; Fig. 5i). In that respect, and considering the limited time available for revision, we did not consider the proposed experiment as a priority to consolidate our key observations and conclusions.

6) I acknowledge the tedious work flow of EM but it will be useful if EM reversed forks could be scored for Figure 4i and 5i.

We note that the role of PARG in modulating RECQ1-mediated fork restart, and thereby the levels of reversed replication forks, has been already shown elsewhere, including key EM observations (Berti et al., NSMB 2013; Zellweger et al., JCB 2015; Ray Chaudhuri et al., MCB 2015). Conversely, we acknowledge that implicating G9a (and thereby H3K9 methylation) in the modulation of fork remodelling/restart is an important novel finding, requiring direct visual support. This is particularly important given that, upon HU-induced fork stalling, H3K9 methylation and fork reversal had been reported as independent events (Gaggioli et al., NCB 2023; see also Reviewer 3, point 6). Hence, we set out to investigate by EM the impact of G9a inhibition on fork reversal upon CPT vs HU treatment, and the role of ADP-ribosylation (PARGi) in this context. The new EM results reported in the revised manuscript (Fig. 5j) provide important clarifications: while G9a inhibition drastically impairs the accumulation of reversed forks upon mild CPT treatment that are permissive for residual fork progression, it has no impact upon HU treatments that stall forks via nucleotide depletion (please note that upon 1h HU treatments, reversed fork degradation is not yet appreciable). As implied by all other evidence in our manuscript, this is most likely to reflect deregulated reversed fork restart by RECQ1 in CPT, which is suppressed if PAR accumulation is restored by PARG inhibition, and is prevented in HU because of nucleotide depletion. These new data provide crucial support to the functional relevance of H3K9 methylation in controlling fork progression and restart upon mild genotoxic treatments, by modulating specific ADP-ribosylation events.

Reviewer #3 (Remarks to the Author):

The manuscript examines the activities of Lamin A/C, histone methylation, and ADP-ribosylation in controlling replication fork dynamics in response to DNA damage. A previous publication found that LaminA/C is important for replication fork stability and restart. That work did not find any defect in fork reversal. The authors explain that limitations in other studies was the use of genetic systems with prolonged inactivation of LaminA/C which could cause cells to adapt or otherwise mask a phenotype. Therefore, they want to examine the consequences of “acute” inactivation. They use both siRNAs and a degron system to inactivate Lamin A/C and use DNA fiber analysis of replication kinetics and EM analysis of replication intermediates to examine replication. They find that inactivating these proteins causes unrestrained fork progression and increased chromosome abnormalities in the presence of DNA damage. Based on localization studies with proximity ligation assays and the similar phenotype of LAP2alpha inactivation, they conclude that nucleoplasmic Lamin A/C is associated with sites of replication and mediates fork slowing and fork stabilization. Furthermore, they find that inactivating Lamin A/C reduces poly-ADP-ribosylation (PAR) near newly synthesized (nascent) DNA but does not alter global cellular levels of PAR. This is important since PAR was previously shown to inhibit the fork restart helicase RECQ1. Thus, they suggest that RECQ1 is not inhibited properly in the absence of Lamin A/C. This causes inappropriate fork restart, a reduction in fork reversal, and the faster fork speeds. Furthermore, they link these phenotypes to a lack of proper H3K9me3 accumulation near the stressed replication forks which prevents proper local chromatin compaction. They conclude that Lamin A modulates local chromatin compaction which then increases PARylation which inhibits RECQ1. These results contrast with the previous studies that reported that Lamin A was required for fork restart and fork protection via regulation of RPA and RAD51. Furthermore, the previous study indicated there was no fork reversal defect in Lamin A-deficient cells (Graziano et al., JBC 2021).

Overall, I found the data interesting but several of the conclusions are not sufficiently justified. I outline some specific concerns below.

1) The localization data is based on a proximity ligation assay between Lamin A/C and EdU. This assay could return a positive result for any abundant nucleoplasmic protein. To ensure there is actual enrichment of Lamin A/C at replication factories, the authors should do an experiment in which EdU is added and then removed for some time to label DNA that is not at replication factories. If there is selective localization of Lamin A/C at replication factories then the PLA signal should no longer be visible if the EdU is no longer near the replication factory.

Binding/proximity to nascent DNA has been assessed in multiple laboratories by imaging approaches (i.e. PLA, or SIRF) or biochemical assays (iPOND, NCC, etc.). While core components of the replisome are expected to show specific binding to nascent DNA, for many replication-associated factors, chromatin modulators and even nucleoskeleton components binding to mature chromatin (i.e. adding a chase period after EdU labelling) is expected and does not challenge the genuine binding of these proteins also to ongoing forks. This is most likely because replicated chromatin retains significant interaction with regulatory factors even well behind replication forks (see for example the nascent/mature binding for most chromatin factors in Alabert et al., NCB 2014, including many that were shown to play important roles in replication). We note, moreover, that – if the PLA signals we detect were simply reflecting accidental proximity to nascent DNA of a particularly abundant protein – we would have observed much higher/abundant PLA signals in the nuclear periphery, where the vast majority of Lamin A/C resides and where its local concentration is reportedly much higher, as confirmed by our own IF images (Fig. 1 and Ext Data Fig. 1). The distribution of the Lamin-EdU PLA signals we detect is instead far more homogenous throughout the nucleus than the Lamin-IF signal, ruling out the artifact proposed by the reviewer (as briefly discussed in the first section of the Results). Importantly, several lines of evidence in our manuscript – further consolidated after revision – confirm that Lamin A/C is playing an important functional role modulating fork progression and restart throughout the nucleus, providing additional indirect support to the interpretation of our PLA results.

2) It is not surprising that the amount of PLA between Lamin A/C and EdU would be decreased when cells are first treated with DNA damaging agents that slow replication since the amount of EdU incorporated would decrease due to slower elongation and decreased origin firing. Therefore, the conclusion that localization decreases in response to replication stress is not justified by the experiment in Figure 1e.

We thank this reviewer for raising this important point (shared with reviewer 1, point 1) and apologize for not sufficiently stressing in the submitted manuscript how we addressed this potential issue. Indeed, PLA signals for proximity to nascent DNA are sensitive to EdU incorporation levels, which are expected to change when cells are treated with genotoxic agents (albeit in very mild doses, as those used in this work). To avoid this issue, we have performed pilot experiments and adapted EdU incorporation times upon treatment, in order to obtain comparable EdU incorporation signals in all conditions and allow reliable PLA measurements. In the revised manuscript, we show these comparable EdU incorporation levels – which were observed with minimal variations throughout our experiments – by displaying a representative PLA experiment in both Ext Data Figs. 1(c) and 4(e), while presenting and discussing in the text how Lamin- or PAR-levels at forks are affected upon specific treatments and/or specific genetic perturbations (Fig. 1e and Fig. 4c, respectively).

3) A key aspect of this report compared to prior studies in the use of “acute” depletion of Lamin A. This is used as one rationale to explain differences in results. Further studies on this point are needed.

a. Does chronic inactivation of Lamin A yield different results in these author’s hands? This could be done by depleting Lamin A with the degron system for multiple days or even a couple of weeks.

b. The AID2 degron system often can yield complete protein degradation in 1-2 hours. The investigators only show how much is depleted at a 24h time point and their experiments were done at that time. Many secondary effects could have happened in 24 hours. Why not do even shorter time points like 2 hours?

We have addressed these concerns by re-investigating Lamin A/C depletion at short (1, 2 or 5h) vs long (1-15 days) time points after 5-Ph-IAA addition, assessing Lamin A/C protein levels and cell cycle distribution in our *mAID2-LMNA* HCT116 cells. Based on these data (reported in Ext Data Fig. 2c-f), we confirmed that a) full protein depletion is not achieved in a few hours, supporting our choice to select 24h as time point for acute Lamin A/C depletion, b) full Lamin A/C depletion is maintained up to 15 days after 5-Ph-IAA addition, without significantly impacting cell cycle distribution in these cells. Importantly, the marked defect in active fork slowing induced by Lamin A/C depletion was clearly observable both at early (1d) and at late (15d) time points (see Ext Data Fig. 2k), excluding that these cells rapidly adapt by promoting Lamin A/C-independent mechanisms of control of fork restart. Although these studies do not exclude that such adaptations to Lamin A/C inactivation may in fact occur in other cellular systems, this evidence - along with new important evidence obtained during revision (e.g. Fig. 5j) - suggests that the different results compared to previous studies mostly reflect the different types of challenges to DNA replication that were investigated. Indeed, while previous studies mostly focused on prolonged fork stalling by HU-induced nucleotide depletion (i.e. a condition that is not permissive for reversed fork restart and residual fork progression; Singh et al., MCB 2013; Graziano et al., JBC 2021), our study investigated the role of Lamin A/C – LAP2 α in modulating fork progression, remodeling and restart upon mild genotoxic treatments that are compatible with residual DNA synthesis (CPT/ETP in nM range). This important point has been clarified in the revised Discussion (page 8), by smoothening and/or removing previous statements on the relevance of acute vs prolonged deficiency (see page 9, first paragraph of the Discussion).

Furthermore, the authors extensively make use of siRNA treatment for 48 hours which they also consider “acute” depletion. It is unclear why some experiments are only performed with the siRNA and some with the degron system. For example, the chromosomal abnormalities in Figure 2 and the unrestrained fork progression analysis and fork reversal analysis in figure 3 are done with siRNA. Are the same results obtained with a short degradation of LMNA protein? The important analysis of ADPr levels is also only done with the siRNA. Why switch between systems? I suggest the authors complete key experiments using both the degron and siRNA. Otherwise the reader is left wondering if the other system yielded different results.

While certain analyses were technically more feasible in specific cellular systems (e.g. Lamin-EdU PLA signal distribution in HCT116 cells, thanks to their round-shape nuclei), we have tried our best to confirm all key findings of the paper in more than one cellular system. Differently from the reviewer’s claim, already in the submitted manuscript unrestrained fork progression was confirmed in two different clones of our *mAID2-LMNA* HCT116 cells, as well as in U2OS cells, with two different genotoxic treatments (CPT/ETP; see Fig. 2a-h). It proved admittedly more challenging to extend to both cell lines more time-consuming approaches, such as EM analysis (where the lab has mostly used U2OS cells and CPT treatments in previous studies), or chromosomal abnormalities by metaphase analysis. Importantly, we have now succeeded extending the ChromStretch analysis also to the *mAID2-LMNA* HCT116 cells, which had proven technically challenging for the first submission and has now yielded important confirmations of our key results (impaired H3K9me3 and H3 density upon Lamin A/C inactivation). Overall, although it was impossible to duplicate all assays in both cellular systems, we feel that all key claims in the manuscript have been confirmed. Moreover, we have included a general statement in the revised Discussion, acknowledging that Lamin/chromatin/ADPr-mediated modulation of replication fork progression is likely to use slightly different regulatory mechanisms in different experimental systems, especially if they display different profiles in terms of chromatin compaction, gene expression and epigenetic marks (page 10).

4) Extended data 2c is used to show that there is no “marked alteration” of cell cycle progression after LMNA degradation. However, 5h after addition of auxin to clone 13 there is over a 10% increase in EdU positive cells and 24h there is a significant number of cells with between 2n and 4n DNA content which are EdU negative.

This point has been addressed by new data included in the revised manuscript, discussed while responding to point 2 here above.

5) I don’t understand how G9a inhibition (and subsequent changes in local chromatin) would affect the level of PAR at replication forks. Does it affect PARP recruitment to forks? Does it affect the amount of CPT-induced DNA damage at forks? Does it alter the activity of PARP1/2? Does it alter the activity or levels of PARG? The previous report by one of these authors found an increase in PARG at forks in response to G9a inhibition and reduction in RECQ1 (Gaggioli et al., NCB 2023). Are those differences underlying these effects?

The reviewer raises several legitimate and important questions here, which admittedly are not answered – if not marginally – by the data included in our manuscript. What we report here is a new axis involved in the control

of RECQ1-mediated fork restart, involving Lamin A/C, the accumulation of H3K9me3 marks and the levels of specific ADP ribosylation events in proximity to replication forks. How specifically H3K9 methylation impacts on local ADP ribosylation levels (e.g. controlling PAR synthesis or degradation) and which specific ADPr targets mediate this effect will clearly require further mechanistic investigation, which we consider beyond the specific focus of this manuscript. Nonetheless, our revision work has extended previous ChromStretch observations, to show that inactivation of the H3K9me3 demethylase KDM3A – which was shown in the submitted manuscript to restore control levels of H3K9me3 at forks upon Lamin A/C inactivation – is also sufficient to restore RECQ1 control and thereby active fork slowing in Lamin A/C-defective cells (Fig. 6g-i). Although these new data do not directly address the mechanistic questions of the reviewer on the links between H3K9 methylation and ADP ribosylation, we feel that this additional evidence further consolidates the main sequence of events underlying Lamin A/C control of RECQ1 fork restart activity, via H3K9 methylation and local ADP ribosylation events at replication factories.

6) The previous paper also showed that G9a inhibition caused fork degradation implying that it did not prevent fork reversal (Gaggioli et al., NCB 2023). How does that fit with the model that G9a promotes reversal and fork slowing?

We thank the reviewer for pointing out this apparent discrepancy, which we have now addressed experimentally. Indeed, previous evidence by the Taneja group showed that HU-induced fork reversal (and the associated degradation) was not prevented by G9a inhibitor. Hence, we set out to investigate by EM the impact of G9a inhibition on fork reversal upon CPT vs HU treatment, and the role of ADP ribosylation (PARGi) in this context. The new EM results reported in the revised manuscript (Fig. 5j) provide important clarifications: while G9a inhibition drastically impairs the accumulation of reversed forks upon mild CPT treatment that are permissive for residual fork progression, it has no impact upon HU treatments that stall forks via nucleotide depletion. As implied by all other evidence in our manuscript, this is most likely to reflect deregulated reversed fork restart by RECQ1 in CPT, which is suppressed if PAR accumulation is restored by PARG inhibition, and is prevented in HU because of nucleotide depletion. While reconciling our new evidence with the previous report from the Taneja lab, these new data provide crucial support to the functional relevance of H3K9 methylation in controlling fork progression and restart upon mild genotoxic treatments, by modulating specific ADP ribosylation events.

7) The authors make conclusions about local chromatin “compaction”. But they do not measure compaction in any experiment. Measurements of a single methylation mark seems insufficient to make conclusions about the chromatin state.

In order to assess chromatin density in parallel to epigenetic marks at single-molecule level, the Taneja lab has further optimized ChromStretch staining conditions to allow for quantitative analysis of H3 density across replicating stretched fibers. These improved experimental conditions revealed that, upon mild CPT treatment, a subset of replication forks in U2OS cells exhibits elevated deposition of H3K9me3 marks, which consistently coincides with increased H3 density (Fig. 6 and Extended Data Fig. 5). Extending this analysis to our second cellular system (*mAID2-LMNA* HCT116 cells) further confirmed our previous observations on the role of Lamin A/C in promoting H3K9me3 accumulation at replication forks under mild genotoxic stress (Extended Data Fig. 5). Moreover, these results show that Lamin A/C-mediated H3K9me3 accumulation is indeed accompanied by detectable chromatin compaction, even under low levels of replication stress.

8) How do the authors reconcile their conclusion in this manuscript that PARG inhibition does not affect fork progression with their previous paper that reported that PARG inactivation causes replication fork slowing (Chaudhuri et al., MCB 2015)?

We thank this reviewer for this important remark, which allows us to clarify this point and avoid possible confusion in the readers. PARG inhibition in the cited manuscript had been achieved by prolonged PARG inactivation and/or by using first-generation PARG inhibitors which are now considered unspecific, which may have led to the detectable effects on fork progression detected in our previous study. This point has now been clarified by including a short comment in the Results section describing these data (page 7).